# Metaplastic Barrett's oesophagus represents reversion to a developmental-like epithelial cell state

Syed Murtuza Baker[1,‡], Aoibheann Mullan[1,*,‡], Rachel E. Jennings[1,2], Karen Piper Hanley[1,3], Yeng Ang[4,5], Claire Palles[6], Neil A. Hanley[1,6,7,§] and Andrew D. Sharrocks[1,§]

## ABSTRACT

In Barrett's oesophagus (BO), the precursor of oesophageal adenocarcinoma, the adult stratified squamous epithelium is replaced by a simple columnar phenotype. This has been considered metaplasia, i.e. the inappropriate conversion from one adult cell type to another. Alternatively, BO could represent reversion to an embryonic-fetal state when the early foregut is initially lined by simple columnar epithelium. Exploration of this hypothesis has been hampered by inadequate molecular details of human oesophageal development. Here, we adopted single-cell transcriptomic and epigenomic approaches to discover and de-code the cell types that constitute the initial primitive columnar, transitory and subsequently stratified lower oesophageal epithelium. Each stage comprises several previously undefined epithelial subpopulations. Importantly, early foregut columnar epithelial cells share core regulatory and gene expression programmes with BO. Among these, HNF4A is identified as a prominent transcriptional regulator that forms the core of a regulatory network in early foregut columnar cells. These regulatory networks are also central to programmes known to be reactivated in BO. Collectively, these data argue that the path to BO involves reactivation of pathways that define primitive embryonic and fetal epithelial cell states.

KEY WORDS: Human embryo, Oesophagus, Barrett's, Oesophageal adenocarcinoma, Gene regulation, Single cell, Chromatin

## INTRODUCTION

Oesophageal adenocarcinoma (OAC) is associated with major mortality and morbidity, with 5-year survival at less than 20% (Coleman et al., 2018). Incidence has risen markedly over recent decades and has been linked strongly to a background of metaplasia, adult-to-adult cell conversion, in the lower third of the oesophagus associated with gastro-oesophageal reflux disease (Smyth et al., 2017). This metaplasia is termed Barrett's oesophagus (BO) and is characterised by replacement of the normal multi-layered (stratified) squamous epithelium by a simple columnar lining that shows unusual intestinal characteristics (Peters et al., 2019). The cellular mechanism for how BO arises has remained unclear. Several theories have been proposed, including transdifferentiation of the stratified squamous epithelium, re-population of the lower oesophagus *in situ* from an altered submucosal stem cell compartment (reviewed by Hayakawa et al., 2021), or, more recently, upward migration of stomach columnar epithelial cells from at or below the gastro-oesophageal junction (GOJ) (Polak et al., 2015; Nowicki-Osuch et al., 2021; Singh et al., 2021). In large part, the challenge of identifying the cell of origin stems from the cells in BO failing to replicate in totality any discrete aspect of the adult gastrointestinal tract. Recently, this led us to hypothesise that BO, and in turn OAC, might actually invoke reversion to a primitive state more reminiscent of embryonic (up to 56 days post-conception) and/or fetal (until birth) development, which primes variable components of misdirected gastrointestinal re-differentiation. We discovered a gene regulatory network (GRN) in BO and OAC centred on the crucial developmental transcription factors (TFs) hepatocyte nuclear factor 4 alpha (HNF4A) and GATA-binding protein 6 (GATA6), neither of which are ordinarily expressed in the adult oesophagus (Rogerson et al., 2019). Indeed, HNF4A is capable of opening chromatin in squamous oesophageal epithelial cells to drive acquisition of a BO-like transcriptional signature (Rogerson et al., 2019) and associated phenotypic changes (Grimaldos Rodriguez et al., 2023). Others have since demonstrated that HNF4A is able to transform adult gastric epithelial cells into a BO-like state (Nowicki-Osuch et al., 2021; Singh et al., 2022).

Understanding whether BO might be a reversal of development has been blocked by rudimentary knowledge of oesophageal development in humans compared to other mammals. Structurally, the oesophagus first forms during the fourth week post-conception as the major part of the foregut endoderm between the buccopharyngeal membrane and the dilation that gives rise to the stomach (reviewed by Zhang et al., 2021). The TF sex determining region Y-box 2 (SOX2) is needed for oesophagus formation and marks the initial simple columnar epithelium (Que et al., 2007; Trisno et al., 2018). By inference from mouse, transition to the multi-layered epithelium is controlled at least in part by the TF TP63, from cells in the basal stem cell compartment of what eventually becomes stratified squamous epithelium (Rosekrans et al., 2015; Zhang et al., 2021). Further molecular details in humans are exceedingly limited. Therefore, in this study we set out to dissect the molecular development of the lower oesophagus at single-cell resolution, and to impute the GRNs and trajectories responsible for the dynamic advance from simple columnar towards stratified squamous epithelium. The findings led us

[1]Faculty of Biology, Medicine & Health, Manchester Academic Health Sciences Centre, University of Manchester, Oxford Road, Manchester M13 9PT, UK. [2]Endocrinology Department, Manchester University NHS Foundation Trust, Grafton Street, Manchester M13 9WU, UK. [3]Wellcome Centre for Cell-Matrix Research, University of Manchester, Oxford Road, Manchester M13 9PT, UK. [4]School of Medical Sciences, Faculty of Biology, Medicine and Health, University of Manchester, Oxford Road, Manchester M13 9PT, UK. [5]GI Science, Salford Royal Hospital, Northen Care Alliance, Stott Lane, Salford M6 8HD, UK. [6]Department of Cancer and Genomic Sciences, College of Medicine & Health, University of Birmingham, Edgbaston, Birmingham B15 2TT, UK. [7]University Hospitals Birmingham NHS Foundation Trust, Birmingham B15 2GW, UK.
*Present address: School of Health and Sports Sciences, Liverpool Hope University, Hope Park, Liverpool L16 9JD, UK.
‡These authors contributed equally to this work

§Authors for correspondence (andrew.d.sharrocks@manchester.ac.uk; neil.hanley@manchester.ac.uk)

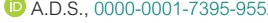 A.D.S., 0000-0001-7395-9552

to undertake complementary studies to explore the relevance to the cellular mechanism underlying BO.

## RESULTS

### Morphology of human oesophageal development

We first wanted to define reliably which time periods would allow us to capture the dynamic epithelial changes in the developing lower oesophagus. We observed that the epithelial layer converts from a simple columnar ('primitive') state, uniformly present at 7-8 weeks post-conception (wpc), to a 'transitory' epithelium captured at 9-12 wpc, which by that point contains regions of several layers with a basal cell compartment and a luminal surface that appears to be ciliated (Fig. 1A; Fig. S1A). A more complex multi-layered epithelium came to predominate ('stratified'; noticeable by 15 wpc), still containing ciliated cells but which became squamous over time and remained that way in the adult (Fig. 1A; Fig. S1A,B). In contrast, the fetal stomach remained lined throughout by simple columnar epithelium. Consequently, a dynamic boundary zone developed in the lower oesophagus leading into the stomach. Initially, this was broad, but narrowed during development. In the adult, it is sharply demarcated as the squamo-columnar junction, commonly also known as the GOJ (Fig. S1B). Having defined when we could represent these different stages and features of human oesophageal development ( primitive, transitory and stratified), we went on to dissect tissue from the lower oesophagus at 7, 9 and 15 wpc to study the individual cell types in molecular detail using a combination of single-nuclear (sn) RNA sequencing (RNA-seq) and assay for transposase-accessible chromatin using sequencing (ATAC-seq) (Fig. 1B).

### Cellular composition of the developing oesophagus

First, we performed snRNA-seq on 2121, 2972 and 4615 nuclei at primitive (7 wpc), transitory (9 wpc) and stratified (15 wpc) stages, respectively. At each time point, we visualised the different cell populations by uniform manifold approximation and projection (UMAP) and identified 8-12 distinct clusters of cells (Fig. 1C-E), which we annotated based on stage, overall gene expression profile and selected marker genes (Fig. 1F-H; Fig. S1C-E; Table S1). For all three time points, some marker genes were predominantly found in a single cluster (e.g. *COL3A1* in cluster 6 and *SMC4* and *KIF23* in cluster 4 at 7 wpc; *EBF2* in cluster 8 at 15 wpc), while other genes distinguished a number of clusters (e.g. *GATA6-AS1* in clusters 1, 4 and 7 and *LRP4* in clusters 2 and 4 in primitive oesophagus at 7 wpc) (Fig. 1F-H; Fig. S1C-E). In several cases, different clusters lacked unique marker genes and instead could only be deconvoluted by combinatorial gene expression. For example, at 15 weeks the two ciliated epithelial subpopulations were distinguished by *ZBBX*, *FANK1*, *DNAAF1*, *AC130456.2*, *KCNE1* and *CROCC2* and discriminated from each other by the presence (cluster 5) or absence (cluster 4) of a wide range of other genes common to other clusters (e.g. *COL12A1* and *SPARC*) (Fig. 1H).

To explore the evolution of cell types across developmental stages, we assembled all the data together into a single UMAP to reveal four broad cell lineages defined by different sets of marker genes: epithelial, mesenchymal, enteric nervous system and blood/vascular (Fig. S2A-C). In keeping with the growing complexity of the oesophagus from its origin as a simple foregut tube, the proportion of epithelial cells declined with advancing fetal age (Fig. S2D). By aggregating across stages, related cell populations could be visualised by the expression of single genes such as *PHOX2A* in neuronal-related cells (Fig. S2E). Similarly, different

epithelial populations could be defined, such as *CFAP73*⁺ ciliated cells, which are present at transitory and stratified stages but not in the primitive epithelium. Other epithelial populations showed overlapping, graded expression of marker genes; for example, combinations of $GATA6^{High}/TP63^{Low}$, $GATA6^{Intermediate\ (Int)}/TP63^{Int}$, and $GATA6^{Low}/TP63^{High}$ expression within and across clusters of developing epithelial cells (Fig. S2A,B,E). $GATA6^{High}/TP63^{Low}$ cells were observed mostly in the primitive and transitory epithelium, but were almost absent in the more complex, stratified epithelia cells by 15 weeks. Conversely, $GATA6^{Low}/TP63^{High}$ cells characterised the basal cells of transitory and stratified epithelium at 9 and 15 weeks (Fig. S2E). The changing composition of the epithelial cell populations as development proceeds reflects the dynamic morphological changes taking place in the epithelium over this time period.

### Development of the oesophageal epithelial layer

To further understand the changes occurring during the development of the epithelial layer, the epithelial populations across the primitive, transitory and stratified stages were combined and re-clustered (Fig. 2A). Three major epithelial cell subtypes were identified: foregut columnar-like, ciliated, and basal stratified cells (Fig. 2A, bottom). Pearson's correlation consistently identified ciliated epithelial cells as the most distinct population (Fig. S3A) with a unique set of marker genes (Fig. 2B) as exemplified by the $CFAP73^{High}$ populations (Fig. 2C). Within the $CFAP73^{Low}$ populations, the hierarchical ordering from the primitive columnar cells through to stratification was consistent with sequential $GATA6^{High}/TP63^{Low}$, $GATA6^{Int}/TP63^{Int}$ and $GATA6^{Low}/TP63^{High}$ expression (Fig. 2C). Amongst primitive populations, RNA velocity analysis implied W4c4 cells were the most rudimentary cells with a clear flow to the other two primitive epithelial clusters (Fig. 2A, W7c4, and Fig. 2D; Fig. S3B) but directional flow between clusters was difficult to discern at later time points. In contrast to the close clustering of all cells at the primitive stage, the cell clusters become separated over time, reflecting the more defined cell types established as a stratified epithelial layer is created (Fig. 2A,D; Fig. S3B). Indeed, a clear switch towards a stratified epithelium was apparent by 15 wpc with an established basal layer, *KRT4*⁺ suprabasal cells and the luminal layer being lined with *FOXJ1*-positive ciliated epithelial cells (Fig. S3C). Broader examination of keratin expression revealed that the establishment of basal cells (*KRT5*⁺/*KRT15*⁺) occurs at the transitory 9-week stage and is maintained at the stratified 15-week stage but suprabasal cells (*KRT4*⁺/*KRT13*⁺) only become established at the later time point (Fig. 2E,F). The epithelial clusters lacked expression of other intestinal marker genes, including *CDX2* (early intestinal), *MUC2* (goblet cells) and *PDX1* (pancreatic) (Fig. S3D). However, the foregut columnar-like clusters found at earlier timepoints shared several gastric markers (including *MUC5AC* and *CLDN18*), particularly at the transitory 9-week stage.

Having begun to discern a likely differentiation pathway, we used a series of marker genes to determine whether any of the developmental epithelial populations are reminiscent of the corresponding adult anatomy, such as cells from the lower oesophagus, upper gastric epithelium as well as the BO metaplastic disease state (as defined by Nowicki-Osuch et al., 2021; Fig. S3E). As expected, the basal stratified cell marker *TP63* was only detected in the normal oesophagus. In normal adult tissue, *GATA6* (foregut columnar maker) was expressed in some gastric epithelial populations, but these cells were unlike those found during development and in BO as they were largely negative for *KLF5* and *HNF4A* expression. *GATA6* was not detected in the normal lower oesophagus. *HNF4A* (another

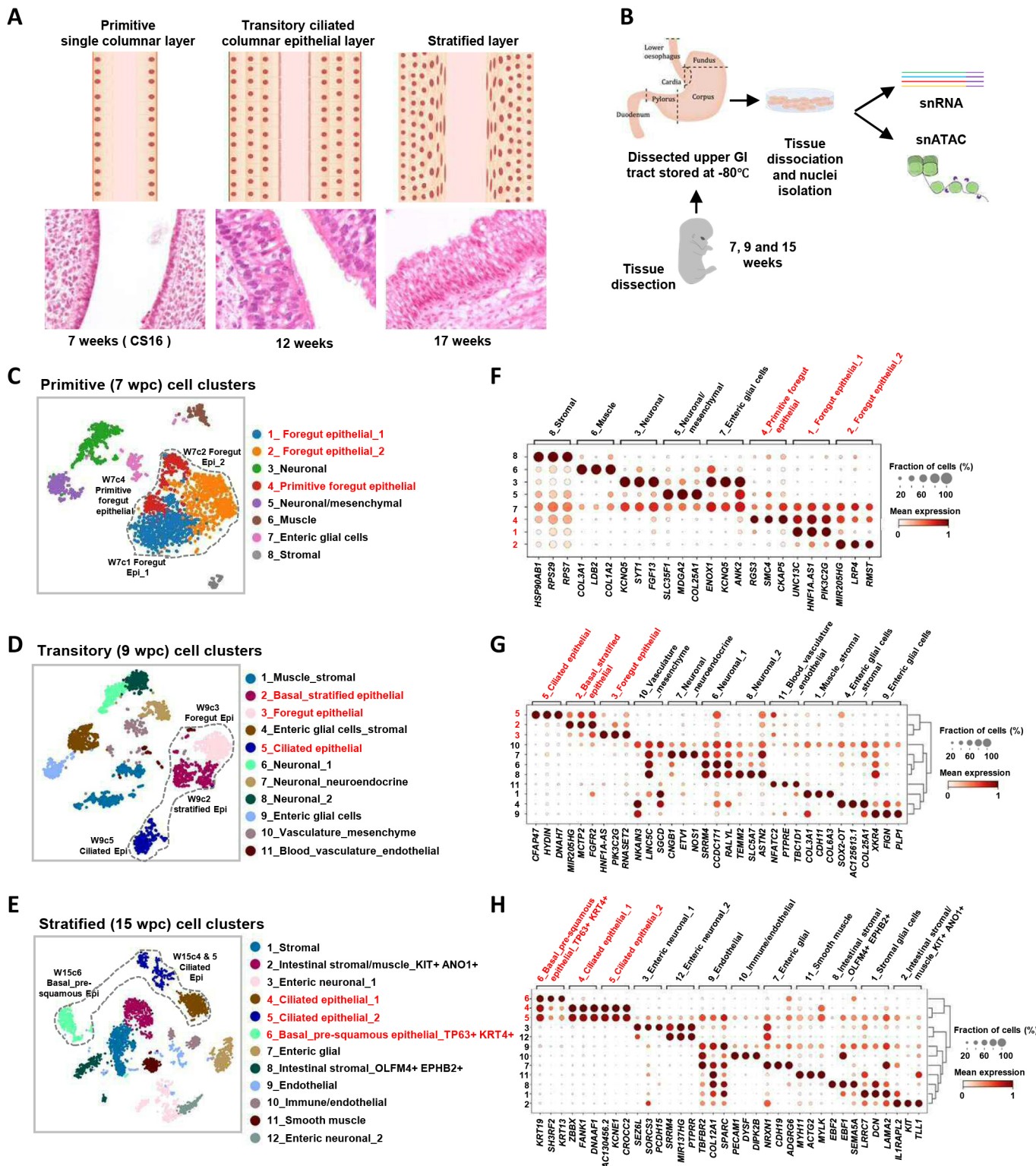

**Fig. 1. snRNA-seq analysis of the developing human oesophagus.** (A) Diagrammatic representations of the structures of the developing oesophagus. Representative H&E-stained sections of the mid oesophagus of each stage are shown below the diagrams. H&E images are also shown in Fig. S1. (B) The overall project workflow from tissue collection to sequencing. (C-E) t-SNE plots of 7 (B), 9 (C) and 15 (D) wpc cell clusters based on snRNA-seq. Major epithelial populations are circled and annotated on the plots. (F-H) Dot plots of the relative average expression of three representative markers for each of the cell clusters found at weeks 7 (F), 9 (G) and 15 (H). The fraction of cells expressing each marker and relative expression levels (column normalised) are represented by the size and intensity, respectively, of each dot.

foreget columnar marker) was only expressed in the BO epithelium. All of the adult clusters lacked cilial markers, such as *CFAP73*, in keeping with the finding that ciliated cells are not generally detected

in the adult oesophagus in North American populations (Scott et al., 2019). Notably, *GATA6*, *HNF4A* and *KLF5* collectively marked cells in BO and this pattern was not replicated amongst any

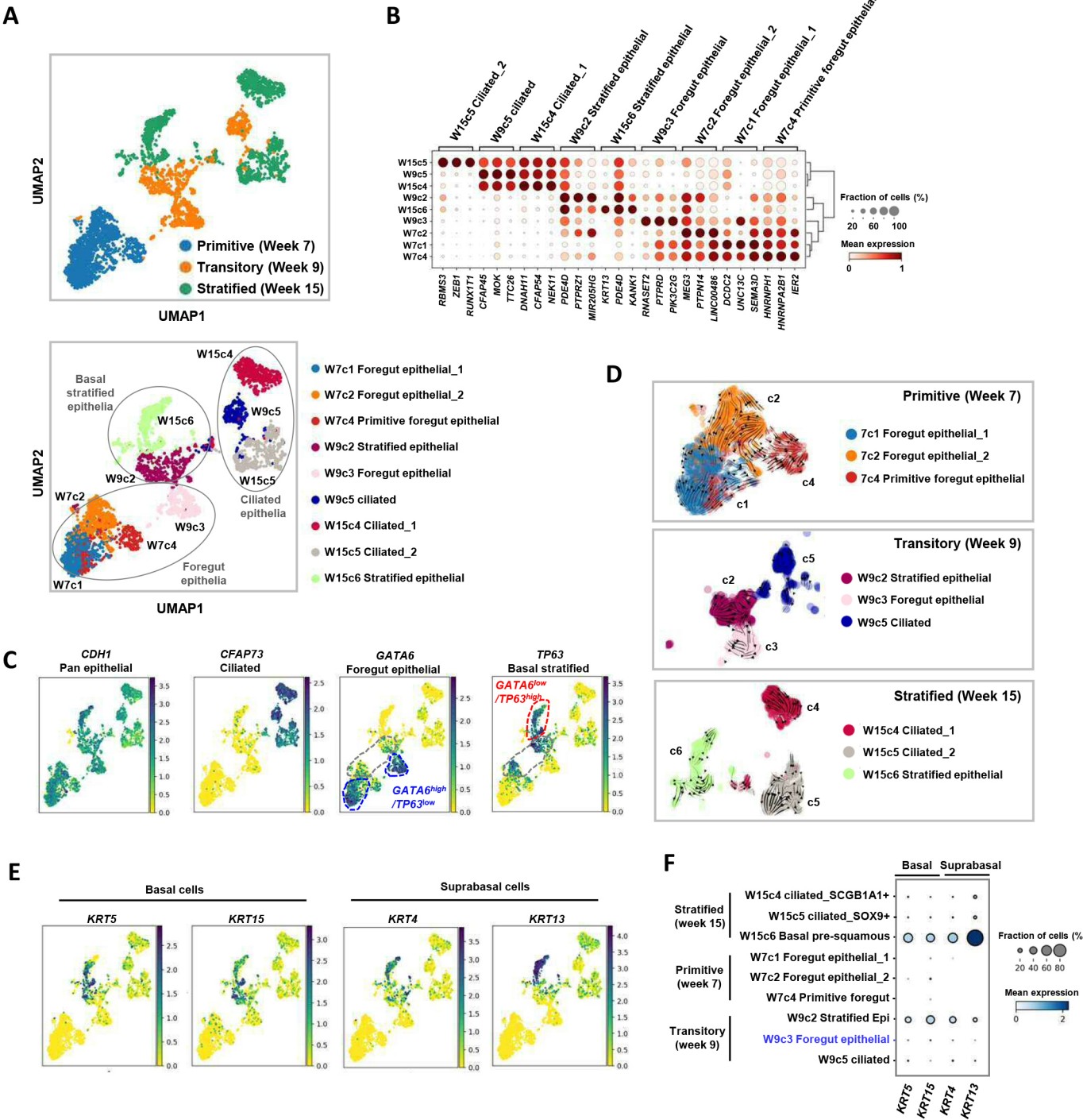

**Fig. 2. Development of the oesophageal epithelial layer.** (A) UMAP of snRNA-seq data from the epithelial cell populations found during embryonic developmental stages at weeks 7, 9 and 15. Clusters are colour-coded according to different epithelial cell types found at each developmental stage (top) or all epithelial cells at each time point (bottom). Major categories of epithelial cells are highlighted. (B) Dot plots of the relative average expression of representative markers for each of the epithelial cell clusters found at different time points. The fraction of cells expressing each marker and relative expression levels (column normalised) are represented by the size and intensity, respectively, of each dot. (C) Marker genes for each of the indicated epithelial cell subtype are shown superimposed on the UMAPs. GATA6$^{high}$/TP63$^{low}$ (blue), GATA6$^{low}$/TP63$^{high}$ (red) and intermediate GATA6$^{int}$/TP63$^{int}$ (grey) cell populations are outlined. (D) RNA velocity (scVelo) analysis of the epithelial cell clusters at each time point superimposed on t-SNE plots. (E,F) Marker genes for basal and suprabasal cells are shown superimposed on the UMAPs (E) or as dot plots of the relative average expression in each cluster (F).

of the adult populations examined but was reminiscent of the primitive and transitory epithelial subpopulations observed during human development.

Collectively, these data demonstrate that during development the epithelial layer undergoes a dynamic transition from a largely simple, primitive foregut in the embryo, through to three major cell types (primitive foregut, ciliated, and basal stratified epithelium) in the progression to the fetal stage, which subsequently resolves into the cells of the stratified epithelial layer, reminiscent of that found in the adult.

## GRNs in the primitive and transitory epithelium

Given the growing evidence of a link between BO and oesophageal epithelial development, we decided to examine GRNs in our embryonic and fetal material to further understand the developmental pathways involved. We focussed on when the epithelial layer undergoes the most dynamic changes and performed snATAC-seq at the primitive (7 wpc) and transitory (9 wpc) stages on 4612 and 481 nuclei, respectively. At the primitive stage, 20 distinct clusters were identified (Fig. S4A, left). All cell states previously identified in clusters in the snRNA-seq data (Fig. 1C) mapped by label transfer to the ATAC-seq-derived clusters except enteric glial cells (Fig. S4A, right). Populations defined singularly by gene expression, for instance muscle cells, could often be resolved as several distinct ATAC-seq clusters (in this case clusters 17, 18, 19 and 20; Fig. S4A,B). This is suggestive of cells with similar gene expression profiles which may be in the process of reorganising their regulatory chromatin landscapes as they transition to different cell states. A similar phenomenon was observed for epithelial cells where four ATAC-seq clusters (1, 7, 8 and 10) mapped to the three populations defined by gene expression, allowing five distinct subpopulations to be discerned when the two modalities were integrated (Fig. 3A, right). These observations prompted us to re-cluster the epithelial populations in the RNA-seq data, which led to the identification of two minor and five major clusters (ca-cg) with distinct sets of marker genes (Fig. S4C-E). $HNF4^+$ (cb) and $TP63^+$ (cc) clusters could be discerned, along with an additional $MUC16^+$ cluster (cd) and a cluster defined by the expression of mitotic cell cycle genes such as $UBE2C$ (ce) (Fig. S4E). The latter is in keeping with our designation of these cells as epithelial progenitors, and is supported by RNA velocity analysis showing a flow from these cycling cells (ce) towards the $HNF4^+$ (cb) and $TP63^+$ (cc) clusters (Fig. S4F,G). To avoid obscuring lineage differentiation, we removed these cycling cells. Subsequent RNA velocity analysis suggested a flow from the intermediate population (cluster ca) to the more defined populations, implicating the ca cluster as a potential progenitor state (Fig. S4H).

We next used the R package ArchR (Granja et al., 2021) to analyse DNA motif enrichments across a trajectory through the newly refined ATAC-seq epithelial clusters at 7 wpc, beginning with the putative progenitor cells in cluster c5* (Fig. 3B). The top variable motifs across the trajectory act as surrogates for TF activity along this trajectory (Fig. 3C). GATA motifs dominated the start of the trajectory, and cells then transitioned through states dominated by HNF4/HNF1, then AP1/GRHL and finally TEAD/TP63 TF activity. These findings corroborate the differentiation path imputed purely from the snRNA-seq data with the transition from $TP63^{Low}$ to $TP63^{Int}$ and then $TP63^{High}$ expression. These cell state transitions were clearly visible by projecting the motif deviation scores for HNF4, GRHL and TP63 family TFs onto the UMAP defined by ATAC-seq (Fig. 3D) and supported by the corresponding imputed snRNA-seq gene expression data for predicted TF activity (Fig. 3E). The identification of $GRHL1$ expression/GRHL activity demarcated a novel transitional epithelial cell population (c4*) between the $HNF4A$/HNF4$^{high}$ and the $TP63$/TP63$^{high}$ cell states. These findings were further substantiated by relative motif enrichments in each of the five newly defined ATAC-seq clusters, with HNF4, GRHL and TP63 motifs again figuring prominently in distinct clusters (Fig. 3F). We also used the Python package SCENIC+ (González-Blas et al., 2023) to derive GRNs in the 7 wpc cell clusters and provided further support for the existence of foregut epithelial cell GRNs driven by members of the HNF4 and GRHL TF families, along with other co-associated GRNs driven by a range of TFs,

including GATA6, FOXA1 and KLF5 (Fig. S5A,B). In keeping with the evolving chromatin landscapes, the changes in expression of prominent regulatory TFs were mirrored by chromatin accessibility changes at their loci as exemplified by $HNF4A$ and $TP63$. The $HNF4A$ promoter and putative intronic enhancer peaks from ATAC-seq foregut epithelial cluster 10 were diminished in the transitional foregut epithelial cluster c7 and virtually absent in the foregut epithelial cell cluster c8 (Fig. 3G, top). Reciprocally, a promoter peak at the $TP63$ locus (giving rise to the short ΔN-TP63 isoform) was absent in ATAC-seq-derived foregut epithelial cluster 10 but emerged in the foregut epithelial cluster c7 and was retained in foregut epithelial cluster c8, when it was accompanied by an additional putative intronic enhancer peak (Fig. 3G, bottom). Collectively, these data identify the key TFs that define different cell states in the primitive epithelium and underline HNF4A and GRHL1 as sequentially acting TFs on the way to dominant TP63 activity as cells progress from primitive $TP63^{Low}$ towards $TP63^{High}$ stratified cell states in the developing lower oesophagus.

We hypothesised that the switch from HNF4A to TP63 activity would be present in transitory cells at 9 wpc, as this stage is still prior to full stratification of the epithelium. Seven distinct clusters were identified at this stage by snATAC-seq, three of which were designated as distinct epithelial subtypes by label transfer from snRNA-seq (Fig. 4A; Fig. S6A). This contrasts with the earlier stage when only closely related primitive foregut epithelial substates were identified. Combining the two modalities yielded four distinct epithelial subpopulations (Fig. 4A, right; Fig. S6B). Ciliated cells remained as a distinct cluster but an additional intermediary group of cells (c2a) emerged with the snATAC-seq profile of primitive foregut epithelial cells and the snRNA-seq profile of basal stratified epithelial cells. A similar phenomenon was observed for other clusters; combining modalities split several of the ATAC-seq clusters, typified by cluster 7, in which the gene expression profiles identified three distinct subclusters, all related to neuronal cell states. This is suggestive of commonly derived cell types with similar epigenetic profiles that are in the process of transitioning to different cell states.

To identify potential developmental trajectories, we first re-clustered the snRNA-seq data associated with RNA-seq-defined basal_stratified (W9c2) or foregut (W9c3) epithelial clusters and applied RNA velocity analysis. Three major clusters were identified, with a newly identified transitional population (analogous to transitional cluster c2a in the ATAC-derived UMAP) that showed directional flow towards either foregut or basal stratified epithelial clusters (Fig. S6C,D). Next, we used ArchR (Granja et al., 2021) to impart trajectories on the overlapping epithelial clusters, and based on RNA velocity analysis each started at the newly defined intermediary state (Fig. 4B). Consistent with being a slightly later developmental stage, fewer cycling cells were apparent (one group constituting the new small subcluster W9c3*) and these contributed little to defining the trajectory analysis (Fig. S6E). We examined the dominant TF motifs as surrogates for TF activity across these trajectories and found that cells at the end of trajectory 1 are characterised by high HNF4 and HNF1 activity whereas those at the end of trajectory 2 contain high TP63 activity (Fig. 4C; Fig. S7A). In both cases, a transitional state was observed hat commonly contained high GRHL activity (Fig. 4C; Fig. S7A) as observed in the transitional state in the epithelial cell populations at 7 weeks (Fig. 3C-E). Plotting TF activity scores on top of the UMAPs (Fig. S7B) or calculating TF activities using DNA-binding motif deviation scores (Fig. 4D) reinforced the trajectory analysis. HNF4 activity was highest in c3 foregut epithelial cells, GRHL in c2a

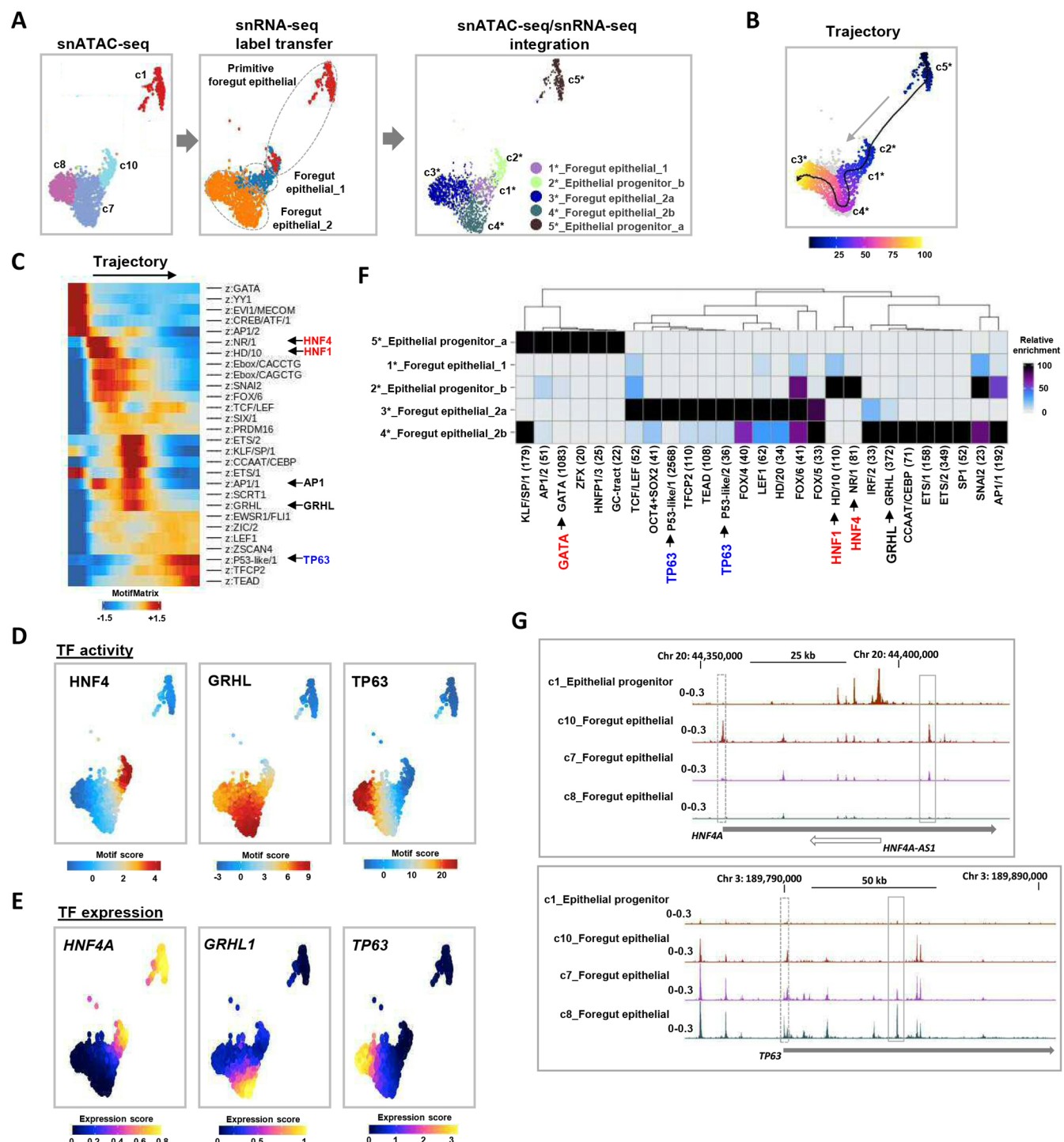

**Fig. 3. Developmental regulatory networks in primitive-stage epithelial cell populations.** (A) UMAPs derived from snATAC-seq data showing the epithelial cell populations in 7-week embryos. Cells are clustered according to snATAC-seq signal (left), annotated based on label transfer from snRNA-seq (middle) and re-clustered based on combined use of snATAC- and snRNA-seq (right). (B) Trajectory analysis of week 7 epithelial cells from clusters c1-5* from the integrated analysis. (C) TF motif deviation score across the trajectory depicted in B. (D) TF-binding motif scores for individual cells projected on the epithelial population ATAC-seq-derived UMAP. (E) Gene expression scores (from gene integration matrix) for the indicated TFs for individual cells projected on the epithelial population ATAC-seq-derived UMAP. (F) Heatmap showing the relative enrichment of the indicated TF-binding motifs in each of the epithelial cell clusters. Scale bar shows enrichment scores in negative $\log_{10}$ of $P$_adj value. Motifs are given family names (Vierstra et al., 2020) and those discussed in the text are highlighted. (G) UCSC browser showing ATAC-seq peaks surrounding the indicated loci and original ATAC-seq-derived cell clusters. Promoter and intergenic/intragenic peaks changing between samples are highlighted by dashed and solid line rectangles, respectively.

transitional cells, and TP63 in c2 basal stratified epithelial cells (Fig. 4D; Fig. S7B). These findings were further strengthened by identifying cluster-specific TF motif enrichments for HNF4, GRHL, TP63 and RFX in each of the four epithelial cell clusters (Fig. S7C). The latter observation was confirmed by plotting motif deviation scores that localised RFX activity to the ciliated epithelial

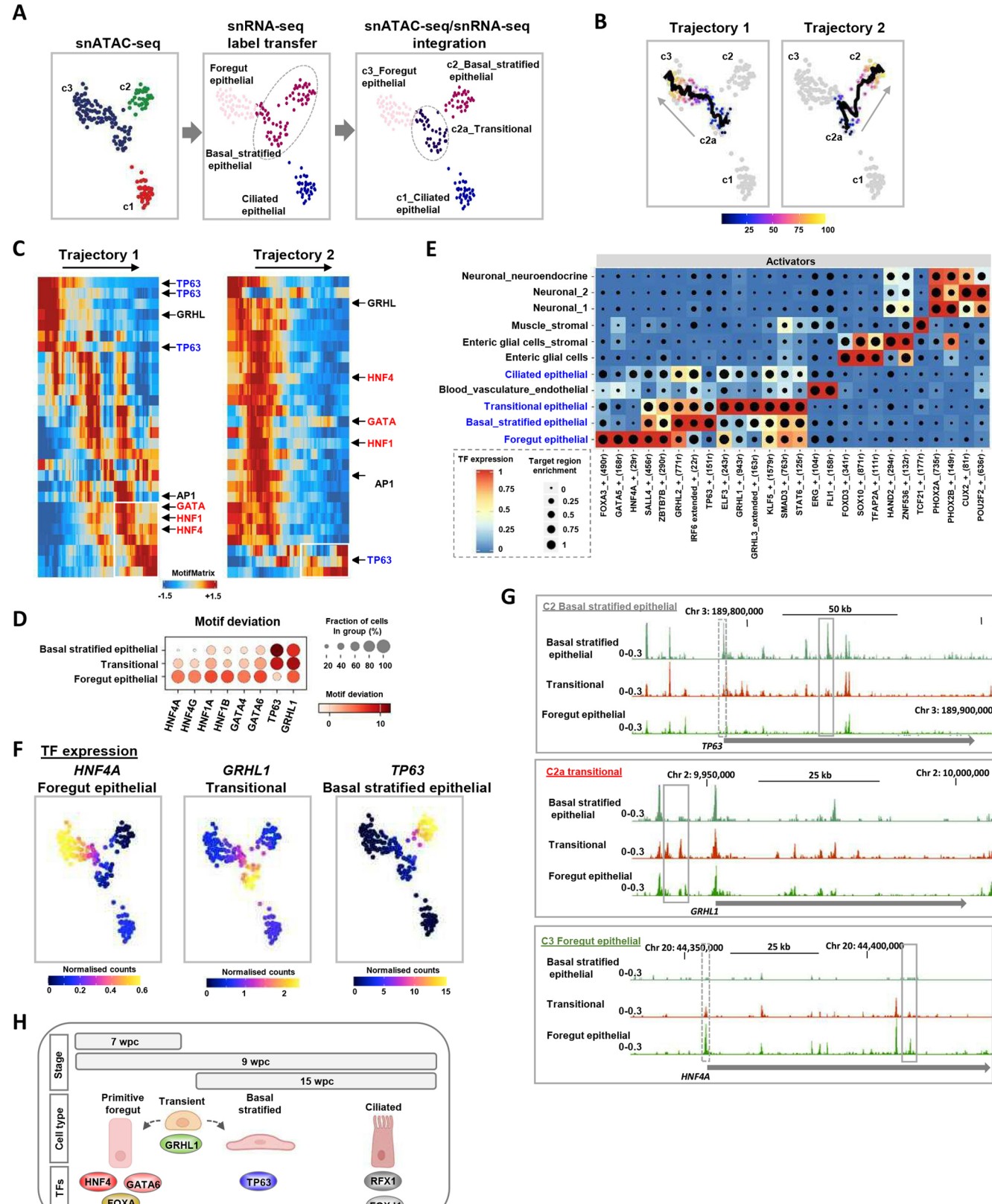

**Fig. 4.** See next page for legend.

cell cluster in keeping with the known role for this TF in specifying ciliated cell fates (Choksi et al., 2014). Moreover, GRN analysis identified regulons for a wider number of regulatory TFs for each cluster, and again uncovered the same categories of regulatory TFs with HNF4 in foregut, GRHL1 in transitional and TP63 in basal stratified epithelial cells (Fig. 4E). Moreover, TF co-regulatory

**Fig. 4. Developmental regulatory networks in the transitory epithelial cells.** (A) UMAPs derived from snATAC-seq data showing the epithelial cell populations in 9-week embryos. Cells are clustered according to snATAC-seq signal (left), annotated based on label transfer from snRNA-seq (middle) and re-clustered based on combined use of snATAC- and snRNA-seq (right). (B) Trajectory analysis of week 9 epithelial cells from clusters c2, c2a and c3. (C) Scaled TF motif deviation score across the trajectories depicted in B. (D) Dot plots showing the deviations in motif scores (indicated by colours) for the indicated DNA-binding motifs in the indicated 9-week epithelial cell clusters (percentage of cells with open motif deviation score indicated by size of dots). (E) GRNs in 9-week cell clusters identified using SCENIC+. The top scoring TF networks are shown (eRegulons correlation coefficient above 0.70 or below −0.65). Cell states are labelled according to label transfer from snRNA-seq. (F) UMAPs showing the expression (gene integration scores from snRNA-seq) of the indicated genes. (G) UCSC browser showing ATAC-seq peaks surrounding the indicated loci and cell clusters. Promoter and intergenic/intragenic peaks changing between samples are highlighted by dashed and solid line rectangles, respectively. (H) Model showing the different epithelial subtypes at each developmental stage, and the dominant TFs defining each cell type.

activity was suggested from correlation analysis, which was particularly marked in the foregut epithelial cluster where HNF4A, GATA5 and FOXA3 regulons coincide (Fig. S7F).

Further analysis showed that expression of members of each TF subfamily broadly follows their predicted activity profiles with high *HNF4A* expression in c3 and high *TP63* expression in c2 (Fig. 4F). *GRHL1* expression was high in the transitional cells located in the c2a cluster between c2 and c3. Similarly, *KLF5* showed highest expression in the transitional population, although this pan-epithelial TF was expressed in all epithelial cell clusters (Fig. S7E). These changes in expression were mirrored by chromatin accessibility changes in the *TP63*, *GRHL1* and *HNF4A* loci where unique peaks emerged in each cluster and, in the case of *HNF4A* and *TP63*, an opening and closing of the promoter region as cells progressed from the transitional state (Fig. 4G).

In summary, these data demonstrate that during development the epithelial layer undergoes a dynamic transition from a largely simple primitive foregut through to the production of three major cell types (primitive foregut, ciliated, and basal stratified epithelium), each defined by dominant regulatory TFs (Fig. 4H). Ultimately, the molecularly defined simple columnar epithelial layer resolves into the stratified epithelial layer that typifies the adult oesophagus. At the earliest stage, closely related primitive foregut epithelial cells represent a mixture of *TP63*-positive and *HNF4A*-positive cells, which resolve via a GRHL1-defined transitional cell state into distinct cell states (driven by HNF4A and TP63) in the transitory stage, and culminating in the retention of TP63[High] cell states and loss of HNF4-positive cells, coincident with the formation of the stratified squamous epithelial layer.

## Relationship between epithelial states in BO and early oesophageal development

Our observation that the primitive epithelial state of early development mimics features of the metaplastic transition of BO led us to deepen our search for shared molecular features. We interrogated published single-cell (sc) RNA-seq data (Fig. 5A; Nowicki-Osuch et al., 2021) and newly generated snATAC-seq data (Fig. 5B) from a non-dysplastic BO sample (Fig. S8). Fourteen cell clusters were identified from 6762 nuclei in the new snATAC-seq data (Fig. 5B). By label transfer from the scRNA-seq data, all of these clusters mapped neatly across with the exception of the small cluster, c10, which was therefore excluded from further consideration (Fig. 5C,D). Amongst the epithelial populations

(clusters c3-9 by snATAC-seq), label transfer pointed to differentiated populations of columnar, endocrine and goblet cells, as well as dynamically changing columnar populations, clearly demarcated by snATAC-seq but with shared or overlapping gene expression profiles (Fig. 5B-D, clusters c5, c6 and c8). By UMAP, these three clusters were flanked left and right by undifferentiated (c7) and differentiated (c4) columnar populations, most likely implying connections through transitional states.

We recently demonstrated that a TF network containing members of the HNF4, HNF1, FOXA and GATA4/6 subfamilies broadly defined the core gene regulatory pathways in BO and that this network is retained in OAC (Fig. 5E; Rogerson et al., 2019; Chen et al., 2020). We hypothesised that the genes encoding these TFs might actually demarcate and be accountable for the different BO epithelial subpopulations. Indeed, their expression profiles overlapped in these populations (Fig. 5F) and, together with activity, as imputed from DNA-binding motif enrichment (Fig. 5G), indicates a regulatory role for the GATA-HNF1-HNF4 axis in the transitional columnar undifferentiated dividing populations. These findings were further underlined by plotting imputed expression and motif deviation scores for TF activity on the snATAC-derived UMAPs where *GATA4*, *HNF1A* and *HNF4A* expression and their corresponding DNA-binding motif deviation scores converged on the transitional undifferentiated epithelial BO populations (Fig. 5H,I). High HNF4A activity was maintained in the differentiated BO epithelial cells. These findings reinforce the idea that the epithelium of BO is itself differentiating and that the GATA-HNF1-HNF4 TF axis plays a key role in this process. This obvious parallel to development led us to return to the epithelial subtypes of the developing oesophagus to see whether we could detect evidence of expression and activity of members of the *HNF4*, *HNF1*, *FOXA* and *GATA4/6* subfamilies. The expression of each gene was predominant in the more primitive cell types, and less apparent in differentiated basal or ciliated cells (Fig. 6A). Co-expression and overlapping imputed binding activity was highest in transitory foregut epithelial cells (cluster c3 at 9 wpc; Fig. 6A-C; Fig. S9A,B). Given these associations, we examined whether we could detect interconnections within the transitory foregut epithelial cell GRNs and found high connectivity among these six TFs (Fig. 6D). This imputed network is reminiscent of longstanding work in mice placing GATA6 upstream from HNF4A in visceral endoderm differentiation (Morrisey et al., 1998).

Based on these clues from key TFs, we looked more globally at potential similarities in expression between BO cell populations and the developmental epithelial clusters. Undifferentiated columnar cells in BO showed strong similarity to epithelial populations at primitive (Fig. S9C) and transitory (Fig. 6E) stages. More specifically, well-recognised markers of BO, for both undifferentiated and differentiated columnar cells (Fig. S9D), were expressed in greatest number in the transitory foregut epithelial cells at 9 wpc (Fig. 6F, cluster W9c3). This strong overlap with foregut epithelial cell cluster W9c3 is exemplified by *MUC5AC*, *TFF3* and *CLDN18* (Fig. 6G), with TFF3, MUC5AC and CLDN18 proteins detected in the lower oesophagus adjacent to the GOJ (Fig. 6H; Figs S9E and S13G). Furthermore, highly connected GRNs centred on the core BO TF network were observed in the transitory foregut epithelial cells (Fig. S10A), including links to many well-known markers of BO (Fig. S10B), and the disease gene ontology terms 'Barrett's oesophagus' and 'intestinal metaplasia' featured prominently within individual and combined regulons for the HNF4A, FOXA3 and GATA5 TFs (Fig. 6I; Fig. S10C-E).

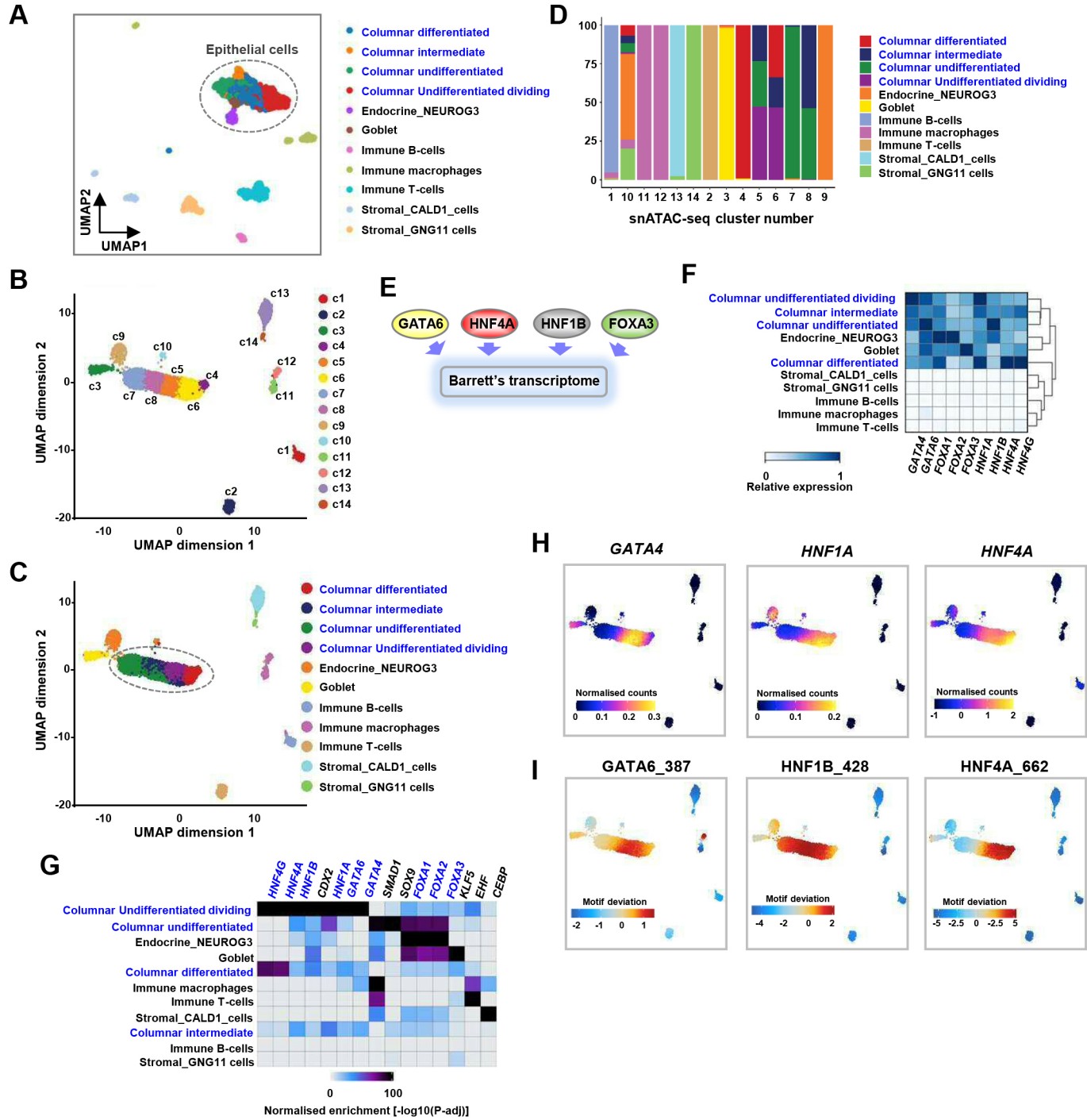

**Fig. 5. Single-cell analysis of Barrett's oesophagus.** (A) UMAP of scRNA-seq data from an individual with non-dysplastic BO. Clusters are annotated as previously (Nowicki-Osuch et al., 2021). Epithelial cell clusters are circled. (B,C) UMAPs of snATAC-seq data from an individual with non-dysplastic BO. Unsupervised clustering is shown (B) and the cell clusters are given identities by label transfer from the scRNA-seq clusters (C). Columnar epithelial cell clusters are circled. (D) Contributions of cells identified by label transfer to each of the ATAC-seq clusters. (E) Diagram of the TF repertoire driving Barrett's cell identity (Rogerson et al., 2019). (F) Heatmap showing the relative expression of the indicated TFs in each of the Barrett's sample clusters. (G) Heatmap showing the relative enrichment of DNA-binding motifs for each of the indicated TFs in each of the Barrett's sample clusters. Enrichment values [−log10(P-adj)] are scaled between 0 and 100. (H) Gene expression scores (from gene integration matrix) for the indicated TFs for individual cells projected on the epithelial population ATAC-seq-derived UMAP. (I) TF-binding motif scores for individual cells projected on the epithelial population ATAC-seq-derived UMAP.

Together, these data identify epithelial cell types in the oesophagus of the developing embryo and fetus that have very similar dominant TF regulatory networks and gene expression profiles to those in BO epithelial cells. These developmental cell types are transient as the oesophagus transitions from a columnar to a stratified epithelium.

## Relationship of epithelial cells in BO to the developing stomach

Comparison of BO to the developing stomach as well as the distal oesophagus is important, given growing evidence implicating gastric epithelial cells as a cell of origin of BO (Polak et al., 2015;

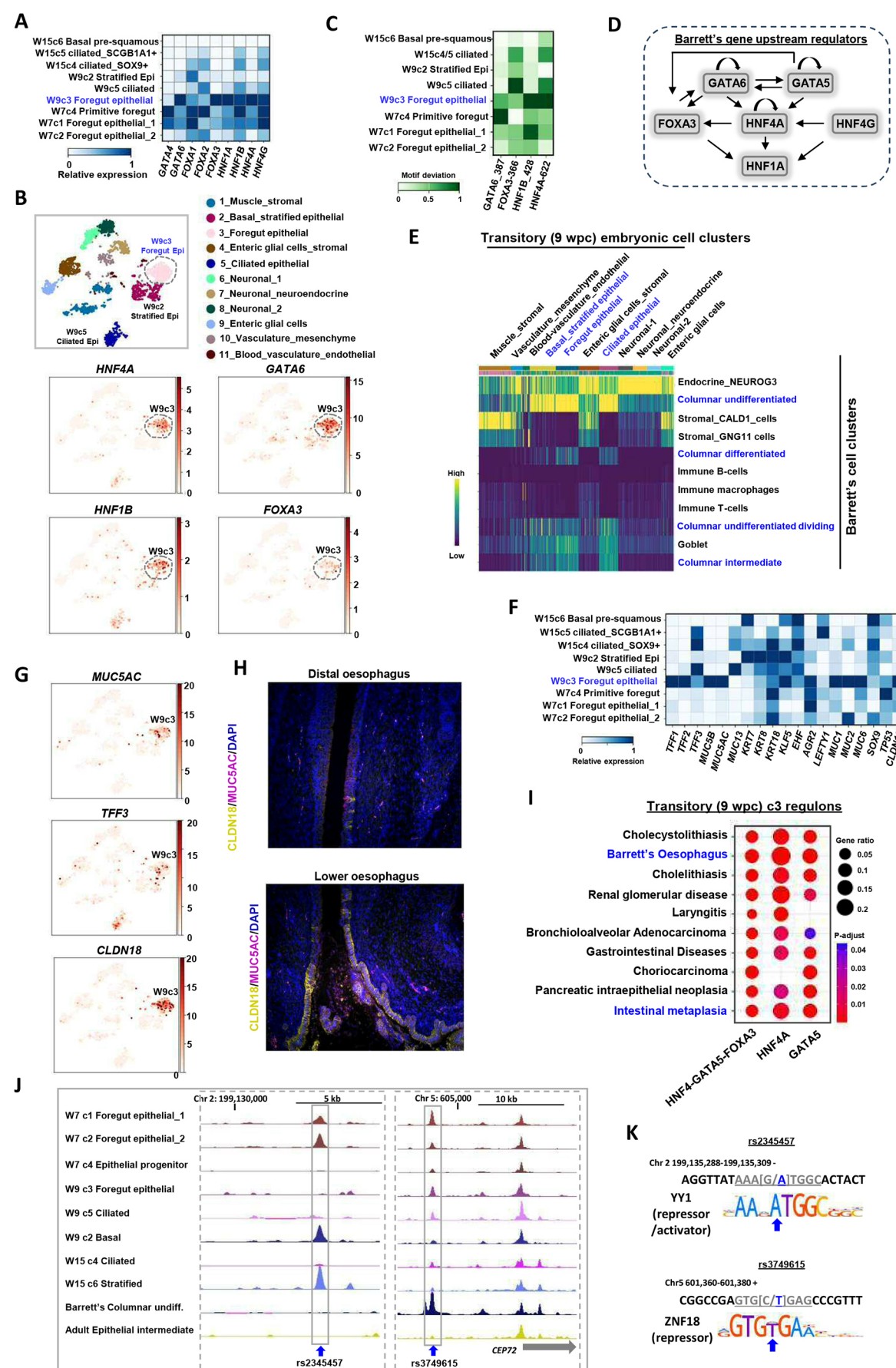

**Fig. 6.** See next page for legend.

**Fig. 6. Barrett's oesophagus resembles a transient developmental epithelial population.** (A) Heatmap showing the relative expression (column normalised) of the indicated TFs in each of the developmental epithelial cell clusters from weeks 7, 9 and 15 (W7, W9 and W15). (B) t-SNE plot showing all embryonic 9-week cell clusters (top) with the expression of the indicated TFs projected on top of these (bottom). (C) TF-binding motif scores (column normalised) in open chromatin regions associated with each of the indicated epithelial cell clusters. (D) Regulatory links derived from the week-9 foregut epithelial cluster GRNs, depicting regulatory interactions between the core TFs. Arrows depict a directional regulatory link. (E) Heatmap showing similarity scores between each cell in the transitory stage 9-week developmental clusters (x-axis) and the corresponding cell types found in Barrett's cell clusters (y-axis). (F) Heatmap showing the relative expression of the indicated Barrett's marker genes in each of the embryonic epithelial cell clusters from weeks 7, 9 and 15 (W7, W9 and W15). (G) t-SNE plot showing all embryonic 9-week cell clusters with the expression of the indicated Barrett's associated marker genes projected on top of these. (H) Immunohistochemistry of MUC5AC and CLDN18 protein expression in week 9 embryos at the GOJ and distal part of the oesophagus. (I) Enriched DisGeNET gene ontology terms for the genes in the combined HNF4A (29r), FOXA3 (490r) and GATA5 (168r), and individual HNF4 and GATA5 regulons in 9-week c2 foregut epithelial cells. (J) UCSC genome browser view of ATAC-seq signals surrounding SNPs rs2345457 (left) and rs3749615 (right) in the indicated clusters of developmental epithelial cells, Barrett's undifferentiated columnar epithelia and adult oesophageal epithelial cells. The arrow below the right track represents the directionality and extent of the *CEP72* gene covered by the views. Peaks containing the SNPs are boxed. (K) Sequences around the significantly associated GWAS SNPs rs2345457 (top) and rs3749615 (bottom). The genomic location and DNA strand is shown above the sequence. Base changes are shown as wild type/risk allele and the risk allele is coloured blue. Logos for the DNA-binding motif of YY1 and ZNF18 are shown and the bases in the corresponding DNA sequence are underlined. Blue arrows indicate the base changed by the risk SNP.

Singh et al., 2021; Nowicki-Osuch et al., 2021). Therefore, we checked for BO-like cells in the developing stomach at a comparable stage to their detection in the oesophagus. We did this by combined snRNA-seq and snATAC-seq analysis ('multiome') for paired lower oesophagus and proximal stomach samples. The oesophageal data validated the previous analysis at 9 wpc, by identifying the same transitory epithelial populations, including foregut-like epithelial cells (clusters 2 and 3), alongside ciliated and basal cells (Fig. S11A-D; Table S2). By motif deviation scores, the same regulatory pathways driven by HNF4, GATA and HNF1 (foregut epithelial, cluster c3) and TP63 (basal, cluster c1) were also imputed (Fig. S11E,F).

Parallel analysis of the stomach identified 12 clusters of cells (Fig. S12A), each with a defining set of marker genes (Fig. S12B,C; Table S2). Epithelial populations included foveolar (*FUT9*+; cluster 2), parietal (*CBLIF*+; cluster 4) and enteroendocrine (*ST18*+; cluster 1). Parietal/chief cells (cluster 3) shared overlapping markers, such as *FGD4*, with cluster 4 (parietal cells) suggesting an intermediary state. The clusters were also distinguished by strong enrichment for a range of imputed TF binding matched by gene expression, exemplified by ELF3/*ELF3* in foveolar cells (cluster 2), ESRRG/*ESRRG* in parietal cells (cluster 4) and RFX6/*RFX6* in enteroendocrine cells (Fig. S12D-F).

We next compared the epithelial populations from the stomach and oesophagus. Oesophageal and gastric cell clusters segregated discretely by anatomical location, apart from some similarity between the gastric parietal cell populations (clusters 3 and 4) and the oesophageal gastric epithelial-like cells (cluster 2) (Fig. S12G). The latter cluster likely represents some inclusion of gastric tissue at the lower boundary of the oesophageal sample. This segregation was reinforced following clustering of the aggregated oesophageal

and gastric oesophageal cell populations (Fig. S13A-C). Importantly, oesophageal-specific (*TFF3*, *CPNE8*, *ZFPM2*) and gastric-specific (*ATP4A*, *VAV3*, *CBLIF*) gene expression could be observed in keeping with their distinct anatomical locations. However, shared genes were identified between oesophageal foregut and gastric foveolar clusters (e.g. *MUC5AC*, *CLDN18*, *MUC1*) suggesting some relationship between these different cell states (Fig. S13D-F). MUC5AC was expressed in the stomach and GOJ. Interestingly, pockets of MUC5AC expression extended up the lower oesophagus (Fig. S13G), consistent with our initial depiction of a broad boundary zone between oesophagus and stomach prior to its resolution later in development into a sharp GOJ (Fig. S1B). More broadly, GATA4 was primarily expressed in gastric epithelial populations and SOX2 in the oesophageal basal-stratified cluster (Fig. S13H), consistent with these two TFs demarcating the GOJ during mouse development (Sankoda et al., 2021).

We next compared oesophageal and gastric developmental populations alongside each other with BO. BO clustered with the transient epithelial populations from the developing oesophagus but with none from the developing stomach; in particular, the more primitive oesophageal foregut population (W7c4, 7 wpc) with columnar undifferentiated BO cells and transitory cells from the oesophagus at 9 wpc (W9c3, W9c2, oesophageal multiome) with the more differentiated BO columnar cells (Fig. S14A). The lack of similarity between BO and any of the gastric populations was mirrored by the more differentiated oesophageal cell types. As a parallel approach, narrowing in on our BO marker genes (Fig. S9D) revealed high differential expression for many of the key genes in the transitory foregut-like oesophageal cells (cluster c3) (Fig. S14B). In contrast, in the stomach only the foveolar cells (cluster 2) showed lower level expression for a subset of BO marker genes (Fig. S14B, C). Genes encoding TFs for the core BO regulatory network were also co-expressed in oesophageal foregut-like epithelial cell clusters, but tended to be distributed individually across different gastric cell populations, implying an inability to function as a TF network (Fig. S14D,E). Taken together, these data point to gene expression in the developing stomach being poorly aligned to that of BO compared to very strong parallels with the transitory oesophageal foregut-like epithelial populations either by genome-scale datasets, the BO marker gene set or known TF networks.

### Barrett's susceptibility loci map to developmental enhancers in the oesophageal epithelial cell populations

Genome-wide association studies (GWASs) have identified 27 sequence variants associated with increased risk of BO and/or OAC, many of which reside in non-coding regions (Schröder et al., 2023). Most of these have not been followed up functionally to identify the causal gene, in part because of a lack of access to RNA-seq and ATAC-seq data from relevant tissue types. While it is plausible some of the variants impact susceptibility to loss of the normal stratified squamous phenotype, others would be predicted to drive acquisition and maintenance of the BO epithelium. Having shown that BO adopts a developmental oesophageal phenotype (rather than metaplasia from one adult cell type to another), we predicted that some BO GWAS non-coding variants would reside in imputed regulatory elements in our newly identified epithelial cell populations from human oesophageal development. Therefore, we overlaid the embryonic/fetal-specific accessible chromatin peaks with the locations of single nucleotide polymorphisms (SNPs) reaching suggestive significance in the BO/OAC GWAS ($P<1\times10^{-5}$) and excluded peaks that were also found in adult epithelial cells. Ten of these SNPs (of which four reached

genome-wide significance; $P<5\times10^{-8}$) overlapped with eight open chromatin peaks in the developmental epithelial cell clusters (Table S3). These ten SNPs marked eight loci associated with BO/OAC [two SNPs were in strong linkage disequilibrium (LD) (r2>0.9) with one other SNP within the set of ten]. Several were in strong LD with previously identified lead SNPs from the BO/OAC GWAS and were associated with regions encoding *SATB2*, *CEP72*, *CFTR* and *MFHAS1* (Table S3; Chen et al., 2022; Schröder et al., 2023). We looked at several in detail; all aligned with open chromatin peaks from different combinations of the developmental clusters, with five of them present in the primitive foregut-like epithelial clusters (Fig. 6J; Fig. S15A). Two of these SNPs were located in peaks unique to developmental epithelial cell clusters (rs2345457 and rs55982826) whereas the other four SNPs were located in peaks also found in the undifferentiated columnar epithelial cells in BO (Fig. 6J; Fig. S15A). Moreover, the SNPs were all predicted to result in reduced activity of the imputed regulatory element, either through the creation of potential repressor-binding sites (rs2345457, rs3749615 and rs1264706; creating YY1, ZNF18 and ZNF708 binding elements) or loss of binding of a transcriptional activator (rs55982826, rs17451754 and rs907183; disrupting KLF5, NFIC and TEAD binding elements, respectively) (Fig. 6K; Fig. S15B).

In summary, our results have uncovered profound similarities in gene expression and in TF regulatory networks between the columnar epithelium of BO and transient subpopulations of oesophageal epithelial cells during development, and that genetic risk variants for BO likely influence the activity of regulatory elements that are transiently active in these epithelial cell populations. In combination, these data support the theory that BO, as a pre-malignant state to OAC, arises directly from reversion of cells in the adult upper GI tract towards the transient epithelial cell states found in the developing human oesophagus.

## DISCUSSION

Towards the end of the first trimester of human development, the developing oesophagus undergoes an intricate conversion from a columnar to a squamous epithelial lining. In its lower portion, this event is intertwined with the creation of the squamocolumnar and GOJ with the stomach. Here, we have used a combined approach of profiling gene expression and chromatin accessibility at the single-cell level to define the cell types that are present in the lower oesophagus and the underlying dominant TFs that specify their phenotypes. Harnessing this information, we discovered that in the disease state, BO, the metaplastic columnar epithelial cells show similarities in their gene expression profiles and regulatory TF networks to a transient epithelial population present in early development before and during the transition to a squamous epithelium. Strikingly, BO is generated by opposite transition from a squamous to columnar epithelium, suggesting that normal development and the mechanism underlying BO might be highly similar processes operating in reverse directions. Indeed, our data build on findings from mouse models of columnar metaplasia that have demonstrated reversion of the squamous epithelium to a fetal developmental-like state before conversion into a columnar epithelium (Vercauteren Drubbel et al., 2021). Our data therefore provide a molecular rationale for this dynamic transition whereby reawakening of the dominant regulatory networks found during development results in re-establishment of an embryonic-like columnar epithelial layer. By extension, this implies a residual epigenetic memory of the developmental state in adult cells in the upper gastrointestinal tract that is susceptible to reactivation following injury to the oesophageal lining. There is likely a strong selective

pressure to attain this state as a potentially protective response to chronic insults and injury. Indeed, such reversion to a fetal-like state appears to be a common event across the intestine as a regenerative response to damage and inflammation (Viragova et al., 2024).

The diversity of cell types in BO are not all developmental in appearance. During development, the oesophageal layer transitions from a series of primitive columnar-like cells in the embryo through to an initial mixed population of ciliated, columnar-like and squamous-like cells. As development progresses, the columnar-like cells, and eventually the ciliated cells, are progressively lost to leave the adult stratified squamous epithelium. The columnar-like cells at primitive stages exhibit imputed activity of TFs from the HNF4, HNF1, GATA and FOXA families. This same network of TFs is re-activated in BO epithelial cells (Rogerson et al., 2019). Commonalities are apparent as HNF4A expression and imputed activity increases as cells become more specified in both development and BO. However, the relative expression of different members of these TFs varies between development and BO. Classical BO marker genes, such as *MUC5AC*, *TFF3* and *CLDN18*, were also unevenly expressed amongst the different epithelial populations both in BO and in development. Further potential similarities are suggested by the recent discovery of *ZMYND10*-positive ciliated cells at the periphery of Barrett's lesions (Gier et al., 2023 preprint), and their relationship with developmental oesophageal ciliated cells (which are also *ZMYND10* positive) warrants further investigation. The lack of complete congruence is unsurprising as any reversion to a developmental state would most likely be imprecise, and the presence of other cell types, such as endocrine cells, could indicate additional aspects of re-differentiation that occur in BO. Moreover, as both the gastric and oesophageal epithelium arise from foregut endoderm, there may be as-yet-uncharacterised endodermal cell types from earlier developmental time points that more closely resemble BO. Indeed, as BO and gastric cardia metaplasia are being increasingly considered as molecularly similar entities (Nowicki-Osuch et al., 2023), such a scenario appears entirely possible. Earlier work in mice would support such a possibility as the core TFs of the regulatory networks we have uncovered, GATA6 and HNF4A, are functionally linked during early visceral endoderm differentiation (Morrisey et al., 1998).

During development, TP63 expression and imputed chromatin-binding activity were detected at the primitive stage in advance of a fully stratified squamous epithelium, suggesting that the regulatory function of TP63 is initiated at a relatively early developmental time point. Only later does TP63 manifest its full activity to drive the gene expression programmes characteristic of the squamous epithelial state. Our detection of RFX TF activity in ciliated epithelial cells marked by *FOXJ1* expression is consistent with both playing known roles in specifying ciliated epithelial cell states (Choksi et al., 2014). In addition to the defined epithelial populations, we also uncovered transitional populations, often expressing mixed columnar and squamous cell markers. It is not clear how these transitions are controlled but at both the early primitive and transitory stages the transitional populations exhibit high GRHL TF family activity, consistent with previous studies that demonstrated a role for these TFs in priming epithelial enhancers for subsequent activation by lineage/cell type-specific TFs (Jacobs et al., 2018).

Several different mechanisms have been proposed for the development of BO based on different cells of origin located in the adult gastric cardia, GOJ and oesophagus (Leedham et al., 2008; Owen et al., 2018; Hu et al., 2007; Quante et al., 2012; Wang et al., 2011; Jiang et al., 2017; Hutchinson et al., 2011; Nowicki-Osuch et al., 2021). While our work does not refute or confirm any of these locations as the definitive source of BO cells, the high similarity of

BO populations to early oesophageal columnar epithelial cells suggests that either the epigenetic memory or the regulatory potential of these developmental cells is maintained in whatever adult cell type undergoes reversion in BO. The acquisition of a fetal-like epigenetic landscape may make cells more susceptible to activation of alternative lineages to generate the additional intestinal-like properties of BO that are not found in the developing oesophagus. It is interesting to note that fetal gastric epithelial cell populations express several of the TFs identified in BO populations, meaning that adult gastric epithelial cells may retain memory of this TF activity and hence be pre-disposed to developmental reversion leading to BO. This would be in line with the gastric origin of BO proposed recently (Nowicki-Osuch et al., 2021). Indeed, it is likely that the magnitude of the epigenetic barriers presented by different adult cell types/states will dictate the most likely cell of origin, which may well represent gastric cell types/ states located close to the GOJ. Moreover, it is unclear how the local microenvironment might affect this developmental reversion. Future work should be directed towards understanding whether extrinsic cues, such as bile acids or the resulting inflammatory environment, influence the activity of the TF networks we have uncovered to trigger regression to an early developmental state. We also find that four out of 27 GWAS significant SNPs associated with susceptibility to BO/OAC map to open chromatin regions representing potential cis-regulatory elements in oesophageal epithelial populations during development. This suggests that susceptibility to BO could have its roots in imperfect assembly and/or function of the oesophageal epithelial layer during development. In this context, microenvironmental cues, such as prolonged exposure to gastric reflux, might influence the activity of these enhancers and more generally may stimulate the activation of TFs such as HNF4A to trigger development of BO.

In summary, our work provides a detailed multiomic cell atlas of the developing human oesophagus and provides insights into the regulatory networks specifying the formation and transition of the epithelial lining. There are, however, limitations. For example, while representative of the developmental events we sought to capture, we have only sampled three developmental time points in our single-cell transcriptome/epigenomic analysis and our spatial analysis is limited to a relatively small number of markers and time points. In addition, although computational predictions are compelling and informative, our data are inherently correlative and do not directly address the cell of origin for BO. Future functional testing in suitable human organoid-based systems, informed by our findings, will be required. Our work complements recent single-cell transcriptomic studies (Yu et al., 2021; Yang et al., 2025), which identified similar ciliated and squamous epithelial cell populations. Our discovery and molecular delineation of earlier epithelial foregut populations at earlier stages are lacking in the Yu et al. and Yang et al. datasets, most likely due to their sampling of later developmental time points (although we cannot exclude technical differences). It was in these more primitive epithelial cell populations that we discovered the developmental TF networks and many gene regulatory aspects shared with BO, reinforcing the theory that reversion to an embryonic/fetal-like state is integral to this disease transition. Ultimately, as BO is a pre-malignant state, extending this knowledge derived from human development may unlock fresh mechanistic understanding for malignant transformation to OAC, which is currently lacking.

## MATERIALS AND METHODS
### Sample dissection
Human embryonic and fetal material was collected under ethical approval from the North West Research Ethics Committee (23/NW/003), with informed consent from all participants and according to the Codes of

Practice of the Human Tissue Authority. Tissue collection took place on our co-located clinical academic campus overseen by our research team ensuring immediate transfer to the laboratory. Embryonic material was staged by the Carnegie classification (O'Rahilly and Müller, 2010). Fetal material (after 56 days post-conception) was staged by foot length and ultrasound assessment. Individual tissues and organs were immediately dissected in cold PBS. In brief, for oesophageal samples the lower third of the oesophagus was dissected to the GOJ, and for gastric samples the upper fundus was dissected from the stomach. All visible adherent mesenchyme was removed under a dissecting microscope. Non-dysplastic BO samples were obtained from biopsies during routine endoscopy. In all cases, tissue was flash-frozen in isopropanol on dry ice.

### Immunohistochemistry
Immunohistochemistry and Haematoxylin and Eosin (H&E) staining was performed in both developmental and adult tissue as described previously (Jennings et al., 2013, 2017), and using the primary antibodies listed in Table S4. Briefly, human embryos and fetuses were fixed within 1 h in 4% paraformaldehyde, processed, and embedded in paraffin wax for orientated sectioning at 5 μm intervals before downstream H&E and immunohistochemical staining.

### snRNA-seq, ATAC-seq and multiome (ATAC and RNA) library construction
For single nucleus RNA-seq and ATAC-seq nuclei were isolated from frozen tissue using the demonstrated 10x Genomics protocol (CG000124). Briefly, tissue was thawed before further dissection. Dissected tissue was incubated in chilled lysis buffer [10 mM Tris-HCl pH 7.4, 3 mM MgCl₂, 10 mM NaCl, 0.1% Tween 20, 0.1% NP40 substitute, 0.01% digitonin and RNase inhibitor (0.2 u/µl)], on ice with gentle pipette and vortex mixing. Incubation time was adjusted according to developmental/adult stage. Cells were strained using 40 and 20 μm filtration, before centrifugation at 500 *g* for 10 min and further washing in wash buffer [10 mM Tris-HCl pH 7.4, 3 mM MgCl₂, 10 mM NaCl, 0.1% Tween 20, 0.1% bovine serum albumin and RNase inhibitor (0.2 u/µl)]. Nuclei were resuspended for downstream RNA and ATAC library preparation using PBS, 0.1% bovine serum albumin, RNase inhibitor and nuclei buffer (10x Genomics Inc.), respectively. For multiome analysis, nuclei were isolated from frozen tissue using the demonstrated 10x Genomics protocol (CG000375), using the reagents outlined above. Nuclei were assessed for quality and quantity using a Countess II automated cell counter. Libraries were constructed using the following kits (10x Genomics): Chromium Next GEM Single Cell 3′ Reagent Kits v.3 (RNA-Seq); Chromium Next GEM Single Cell ATAC Reagent kit v.1.1 (ATAC-Seq); Chromium Next GEM Single Cell Multiome ATAC+ Gene Expression kit (multiome analysis). Libraries were processed on the Illumina NextSeq 500 platform.

### snRNA-seq analysis
#### Read mapping to genome
The sequence files generated from the sequencer were processed using the 10x Genomics custom pipeline Cell Ranger v.3.0.1. This pipeline generated fastq files which are then aligned to the hg38 custom genome with all the default parameters of Cell Ranger. The pipeline then identified the barcodes associated with cells and counted unique molecular identifiers (UMIs) mapped to each cell. Cell Ranger uses STAR aligner to align reads to the genome so it discards all the counts mapping to multiple loci during counting. The uniquely mapped UMI counts are reported in a gene by cell count matrix represented as sparse matrix format. We aggregated all QC-filtered cells in the different embryonic time points to generate the aggregated dataset.

#### Cell quality filtering
Low-quality cells were removed from the dataset to ensure that the technical noise did not affect the downstream analysis. We used three parameters for cell quality evaluation: the number of UMIs per cell barcode (library size), the number of genes per cell barcode and the proportion of UMIs that were mapped to mitochondrial genes. Cells that had lower UMI counts than one median absolute deviation (MAD) for the first two metrics and cells with

higher proportion of reads mapped to mitochondrial genes with a cutoff of two MADs were filtered out. We then created violin plots for these three metrics to see whether there were cells that had outlier distributions, which can indicate doublets or multiplets of cells. We removed outlier cells that had total read counts more than 50,000 as potential doublets. After these filtering steps, 9708 cells (2121 cells from week 7; 2972 cells from week 9; 4615 cells from week 15) remained for downstream analysis.

### Gene filtering and normalisation

Genes with average UMI counts per cell below 0.01 were filtered out when working with individual time points as we assumed that these low-abundance genes would not give much information and were unreliable for downstream statistical analysis (Bourgon et al., 2010). In order to account for the various sequencing depth of each cell, we normalised the raw counts using the deconvolution-based method (Lun et al., 2016). In this method, counts from many cells were pooled together to circumvent the issue of higher number of zeros that are common in single-cell RNA-seq data. Each cell was then divided by a pool-based size factor that was deconvoluted from the pool of cells and then multiplied by a million. These normalised data were then log-transformed with a pseudo-count of 1 added to all counts.

### Visualisation and clustering

The first step for visualisation and clustering was to identify the highly variable genes (HVGs). We first decomposed the variance of each gene expression values into technical and biological components using the 'modelGeneVar' function from scran and identified the genes for which biological components were >0.5 and with false discovery rate (FDR) value <0.05. We called these genes HVGs. These HVGs were then used to reduce the dimensions of the dataset using principal component analysis (PCA). The dimensions of dataset were further reduced to 2D using t-distributed stochastic neighbour embedding (t-SNE) or UMAP, with the first 14 components of the PCA given as input.

The cells at each developmental time point were grouped into their putative clusters using the dynamic tree cut method (Langfelder et al., 2008). Instead of choosing a fixed cut-point for the tree, the dynamic tree cut method identifies the branch cutting point of a dendrogram based on the underlying data and combines the advantage of both hierarchical and K-medoid clustering approach.

We generated an aggregated dataset by combining all cells from the three time points. We started with running Spaceranger aggr on all four samples without any normalisation or down-sampling of reads (one 7 week sample was subsequently dropped due to poor quality). We then conducted the quality control steps in R by first estimating the proportion of ambient contamination genes using DropletUtils. We then identified low-quality cells by looking at median absolute deviation across three parameters – library size, number of genes expressed and mitochondrial proportion – separately for each sample. We applied slightly relaxed thresholds (2.5 MAD for library size and 3 MAD for number of genes detected and mitochondrial proportion) for the aggregated data compared to our individual sample analysis because we wanted to increase signal-to-noise ratio in a heterogeneous population by increasing cell numbers. Furthermore, we removed all cells that had reads higher than 99.5th quantile as potential doublets. For week 7, we set the threshold manually to 25,000 reads as it originally had lower sequencing depth and cells higher than 25,000 reads were clearly outliers. After these filtering steps, we were left with W15-4615, W09-2972 and W07-2121 cells. We then exported these quality-filtered cells to Python and conducted further downstream analysis in Scanpy (Wolf et al., 2018) by first applying PCA on HVGs and then calculating neighbours using 40 components of these PCA. We used these neighbours to calculate Leiden clustering and UMAP for visualisation.

For epithelial cell aggregation, we applied a slightly different approach. For this, we first selected all clusters that we manually annotated as epithelial cells in each individual sample analysis. As the marker gene signatures might vary across the time points, this approach ensured that epithelial cells were annotated at higher resolution. We aggregated these cell counts and converted them into AnnData. We then used Scanpy for further downstream analysis. For this, we calculated PCA after determining HGVs. We then passed 40 PCA components to calculate neighbours using Scanpy's 'neighbour' function. Similarly to our 'all cell' aggregation approach, we

used these neighbours to calculate the Leiden clustering and UMAP of all epithelial cells. We integrated published datasets with our aggregated samples using Harmony for batch correction across samples, followed by UMAP projection and clustering based on the Harmony-adjusted PCA (Korsunsky et al., 2019).

### Identification of marker genes

To identify the marker genes for a cluster, we compared that cluster with all other clusters. We then reported the genes that were differentially expressed in that cluster as the marker for the cluster. We used the 'rank_gene_groups' function from Scanpy package and conducted the Wilcoxon test to find these marker genes. These marker genes were then used to annotate the cell types of a cluster.

### RNA velocity

We applied scVelo (Bergen et al., 2020) to identify transient cellular dynamics in each of our developmental stages. To calculate the spliced to unspliced ratio for each gene, we used RNA velocity's command line tool, velocyto 10x, with default parameters and used scVelo to compute velocity based on the spliced to unspliced ratios.

### snATAC-seq analysis

We used cellranger-atac 2.0.0 to map fastq files to the hg38 genome. In summary, cellranger-atac trims primer sequences from the reads and maps these trimmed reads to the genome using modified version of BWA-mem (Li, 2013 preprint). After removing the PCR duplicates, it then uses a similar algorithm to ZINBA (Rashid et al., 2011) to call peaks. Cellranger-atac then uses reads mapping to these peaks to identify signal from noise and uses it to call cells from background.

For our downstream analysis, we used ArchR tool (Granja et al., 2021). After filtering out the doublets, we did cell filtering based on the transcription start site (TSS) enrichment score and minimum fragments in a cell. Cells having a TSS enrichment score of 4 and a minimum of 1000 fragments passed through our filter. After the QC, we had 6762 cells from Barrett's, 1658 cells from the normal adult oesophagus, 781 cells from week 15, 4688 cells from week 7 and 481 cells from week 9 embryonic/fetal oesophagus. We used latent semantic indexing (LSI) for dimensionality reduction and used graph-based clustering to cluster the cells. Labels from cell-type annotation in snRNA-seq were transferred to snATAC-seq to annotate the clusters with addGeneIntegrationMatrix, which uses Seurat's CCA method to do the label transfer. Peaks on these cell types were then called using MACS2 (Liu, 2014) using the parameter '–call-summits –keep-dup all –nomodel –nolambda –shift -75 –extsize 150 -q 0.1'. After identifying the marker peaks for each cell type, they were then used to do motif enrichment. First, we used the 'getMarkerFeatures' function from ArchR to identify marker peaks. This function compares each peak in a cell group against its own background group of cells to determine whether the peaks have significantly higher accessibility. To create this background for a group of cells, ArchR first identifies the nearest neighbour cells in the multidimensional space that are not part of this group. Peaks that are enriched in a cluster with log-fold change more than 1 against its background group of cells with FDR≤0.01 are annotated as marker peaks for the cluster. The 'peakAnnoEnrichment' function was then used to analyse these groups of marker peaks for enriched binding sites of TFs. We used Cis-BP, a catalogue of direct and inferred sequence binding preferences as the motifset database. TF-binding motifs were grouped according to similarity and annotated based on their (sub)families for concise annotation of figures (Vierstra et al., 2020).

### GWAS analysis with epithelial clusters

We first sub-setted all the epithelial cells from our 'all aggregated' dataset. We then used MACS2 (Liu, 2014) to call peaks from the snATAC-seq data on these epithelial cells. This gave a single peak set for all the epithelial cells. We used this peak set for all our differential accessibility (DA) tests in this section. For DA, we first grouped the cells into two groups: all the developmental and BO epithelial cells constituted group-1 and all the normal adult epithelial cells constituted group-2. We then identified the peaks that were differentially accessible in group-1 compared to group-2.

ArchR calls peaks using MACS2 and then uses an iterative merging strategy to merge the overlapping peaks into a single larger peak. ArchR also uses a fixed width of 501 bp for each peak. We used the 'getMarkerFeatures' function from ArchR to calculate the differential accessibility. As mentioned earlier, ArchR uses a bias-matched group of background cells to compare and calculate the DA. For statistical significance, we used the Wilcoxon test, which tests whether the distribution of peak opening across two groups are significantly different. We first used a threshold of mean difference more than 0.03 between group-1 and group-2 or FDR less than 0.01 to identify the initial peaks. This mean difference was calculated as the differences of the mean of peak accessibility distribution for a peak in two groups. Peaks passing this threshold were then further filtered to keep the peaks that had log fold change of more than 2.75 in group-1 compared to group-2. Peaks passing these thresholds were then overlapped with the GWAS SNPs and we removed the peaks that did not overlap with a SNP. For selecting the SNPs, we only chose those that had a $P$-value less than $1 \times 10^{-5}$ in the most recent OAC/BO GWAS meta-analysis summary statistics ($n$=5563; Schröder et al., 2023). We lifted over the SNP positions from build 37 to build 38 for comparison with our snATAC-seq data. We considered a SNP to be overlapping if it was located within group-1 peaks and differentially accessible compared to normal adult epithelial cells. These overlapping peaks were then checked against their peak score, which was calculated by MACS2 during peak calling as $-\log_{10}$(qvalue) where q values show the significance of the peak. Any peak that had a score lower than 25 were filtered out. Finally, we filtered out any peaks that had a score lower than 10 in the developmental cells only. This ensured that we did not have peaks that were only high in Barett's without also being detected confidently in developmental cells. For this last filtering step, we used peaks that were only called on the developmental cells only.

## Multiome data analysis

We used cellranger-arc 2.0.2 to pre-process our single cell multiome (ATAC and gene expression) sequencing data, which involved performing alignment of the fastq files to hg38 genome, filtering, counting the barcodes, peak calling and counting of both gene expression (GEX) and ATAC modalities. For downstream analysis, we again used ArchR (Granja et al., 2021) as our core multiome analysis framework. We assessed the quality of the cells based on both modalities and filtered out the cells that had TSS enrichment ratios lower than 4 or had mapped ATAC-seq fragments lower than 1000 reads. We also used the ATAC modality to identify potential doublet cells. To infer these doublet cells, ArchR first synthesises *in silico* doublets by mixing reads from thousands of different combinations of cells. Cells matching the profile of these synthetic doublets are marked as potential doublet cells which are then filtered out. We then used the GEX modality to additionally filter out cells that had <500 reads mapped to them or cells that had higher than 5% of mitochondrial reads. After these filtering steps, we retained 2633 cells in gastric and 1240 cells in oesophagus multiomes. We then applied iterative LSI both for individual modalities as well as for joint modalities where ATAC and GEX were joined using a weighted nearest neighbour algorithm (Hao et al., 2021). UMAP (Becht et al., 2018) plots were then generated by applying the algorithm individually on the LSI of ATAC modality, the GEX modality and the combined modality (GEX and ATAC). We then used graph clustering on these three LSI reduced dimensions to get clusters for the snRNA-seq modality, the snATAC-seq modality and for the joint modality. In addition to the gene expression that we obtained from GEX, we also calculated the gene score for the ATAC modality, which is calculated by looking at the mapped reads located 100 kb either side of the gene start. Both the gene expression and gene score were used to annotate the clusters. We use MACS2 (Zhang et al., 2008) to call peaks for each of the ATAC-derived clusters. For our analysis, we used the joint modality clusters for both the datasets. We used a fixed width of 501 bp for the peaks to facilitate downstream computational analysis without needing to normalise for peak length. We ran the Wilcoxon test to identify marker peaks for each cluster from this peakset and performed motif enrichment analysis, as described above for snATAC-seq data, on these marker peaks. We also calculated the motif deviation for each of the motifs, which explains how the accessibility of a motif deviates in a cell from its average accessibility across all cells. We

then conducted TF footprinting to predict the precise binding location of a TF using ArchR (Granja et al., 2021). This combines the Tn5 insertions across many instances of a predicted TF-binding site to address the requirements of higher sequencing depth for identifying a footprint.

We also inferred developmental trajectories using ArchR (Granja et al., 2021), which orders cells across a lower N-dimensional subspace based on a user-defined trajectory backbone.

## Inferring GRNs using SCENIC+

We used SCENIC+ (v.1.0.1.dev4+ge4bdd9f; González-Blas et al., 2023) to infer GRNs using both snRNA-seq and snATAC-seq. SCENIC+ links candidate enhancer regions with binding motifs to their candidate target genes. As input, SCENIC+ uses the snRNA-seq, the cisTopic object and the Topic modelling of the chromatin peaks and the motif enrichment results. For the non-multiome data (scRNA-seq and snATAC-seq not measured in the same cell), we created a pseudo-multiome by generating metacells. These metacells were created by randomly sampling ten cells from each modality (snRNA and snATAC) and taking the average of the counts within these cells. For each of the developmental time points, we first extracted the cell metadata and consensus regions from our ArchR objects. We then created a cisTopic object using this information along with the fragments from the raw fragment file. Next, we applied topic modelling to group the co-accessible open chromatin regions into multiple topics. We used four different quality metrics to choose the optimal number of topics. We then used pyCisTopic (v.1.0a0) to predict the candidate enhancer regions. Next, we used the wrapper of pycistarget within SCENIC+ to identify the motifs that were enriched in regions that were differentially accessible for each cell type and were on the candidate enhancer region. For this motif enrichment, we used a pre-computed motif-score database. Finally, we ran SCENIC+ to infer the GRNs.

## Acknowledgements

We are very grateful to everyone who consented to take part in our research programme and for the assistance of research nurses and clinical colleagues at the Manchester University NHS Foundation Trust. We thank Andy Hayes of the Genomic Technologies Core Facility at the University of Manchester. We thank Johannes Schumacher, Rebecca Fitzgerald and Stuart MacGregor for sharing the GWAS summary statistics and also Karol Nowicki-Osuch and Rebecca Fitzgerald for sharing scRNA-seq data from BO and upper GI tissues.

## Competing interests

The authors declare no competing or financial interests.

## Author contributions

Conceptualization: R.E.J., K.P.H., N.A.H., A.D.S.; Formal analysis: S.M.B., A.M., C.P., A.D.S.; Funding acquisition: A.D.S., N.A.H.; Investigation: S.M.B., A.M.; Project administration: N.A.H., A.D.S.; Resources: R.E.J., Y.A., C.P.; Software: S.M.B.; Supervision: K.P.H., N.A.H., A.D.S.; Writing – original draft: S.M.B., R.E.J., N.A.H., A.D.S.; Writing – review & editing: S.M.B., R.E.J., K.P.H., Y.A., C.P., N.A.H., A.D.S.

## Funding

The work was supported by a Medical Research Council Human Cell Atlas project grant (MR/S036121/1). R.E.J. was a Medical Research Council clinical research training fellow. Open Access funding provided by the Medical Research Council. Deposited in PMC for immediate release.

## Data and resource availability

The single-cell RNA-seq, ATAC-seq and multiome (combined ATAC- and RNA-seq) data have been deposited in ArrayExpress: E-MTAB-14127 (snRNA-seq), E-MTAB-14128 (snATAC-seq) and E-MTAB-14129 (multiome). Genome browser tracks are available that summarise the pseudo-bulk populations for the cell clusters derived from our snATAC-seq data: epithelial cell populations from snATAC-seq at three time points (https://genome-euro.ucsc.edu/s/galib36/Esophagus_epithelial_cells), all cell populations from the gastric multiome (https://genome-euro.ucsc.edu/s/galib36/multiome_gastric) and all cell populations from the oesophageal multiome (https://genome-euro.ucsc.edu/s/galib36/Oesophagus_15W_multiome). Interactive apps are available to explore our snRNA-seq data: all cell populations from snRNA-seq at three time points (https://cellxgene.bmh.manchester.ac.uk/OES_all/), all cell populations from oesophageal multiome (https://cellxgene.bmh.manchester.ac.uk/multiome_oes/) and all cell populations from gastric multiome (https://cellxgene.bmh.manchester.ac.uk/multiome_gastric/). The following

codes are freely available: https://github.com/galib36/developing-human-oesophagus. All other relevant data and details of resources can be found within the article and its supplementary information.

## The people behind the papers

This article has an associated 'The people behind the papers' interview with some of the authors.

## Peer review history

The peer review history is available online at https://journals.biologists.com/dev/lookup/doi/10.1242/dev.204735.reviewer-comments.pdf

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
