## [Peer Review File · Development (Cambridge, England)]

Metaplastic Barrett's oesophagus represents reversion to a developmental-like epithelial cell state

Syed Murtuza Baker, Aoibheann Mullan, Rachel E. Jennings, Karen Piper Hanley, Yeng Ang, Claire Palles, Neil A. Hanley and Andrew D. Sharrocks
DOI: 10.1242/dev.204735

Editor: James M Wells

Review timeline

Original submission:	20 February 2025
Editorial decision:	31 March 2025
First revision received:	11 September 2025
Editorial decision:	2 October 2025
Second revision received:	6 October 2025
Accepted:	13 October 2025

Original submission

First decision letter

MS ID#: dev.204735

MS Title: The metaplastic precursor state to oesophageal adenocarcinoma represents reversion to a transient epithelial cell state in the developing oesophagus

Authors: Syed Baker; Aoibheann Mullan; Rachel Jennings; Karen Piper-Hanley; Yeng Ang; Claire Palles; Neil Hanley; Andrew David Sharrocks
Article Type: Research Article

Dear Dr Sharrocks,

I have now received all the referees' reports on the above manuscript, and have reached a decision. The referees' comments are appended below, or you can access them online: please go to:

As you will see, the referees express considerable interest in your work, but have some significant criticisms and recommend a substantial revision of your manuscript before we can consider publication. If you are able to revise the manuscript along the lines suggested, which may involve further experiments, I will be happy receive a revised version of the manuscript. Your revised paper will be re-reviewed by one or more of the original referees, and acceptance of your manuscript will depend on your addressing satisfactorily the reviewers' major concerns. Please also note that Development will normally permit only one round of major revision. If it would be helpful, you are welcome to contact us to discuss your revision in greater detail. Please send us a point-by-point response indicating your plans for addressing the referees' comments, and we will look over this and provide further guidance.

Please attend to all of the reviewers' comments and ensure that you clearly highlight all changes made in the revised manuscript. Please avoid using 'Tracked changes' in Word files as these are lost in PDF conversion. I should be grateful if you would also provide a point-by-point response detailing how you have dealt with the points raised by the reviewers in the 'Response to Reviewers' box. If you do not agree with any of their criticisms or suggestions please explain clearly why this is so.

Reviewer 1

Baker et al. aim to understand the fetal nature of Barrett's oesophagus (BO) metaplasia and how it resembles a reversion of oesophageal epithelium to a primitive fetal oesophagus program. They utilize snRNA-seq and snATAC-seq to characterize human lower oesophageal development at 3 fetal timepoints, termed primitive (7 wk), transitory (9 wk), and stratified (15 wk) to and gain insight into transition from simple columnar to squamous epithelium and potential GRNs involved that could be perturbed in BO. They focus on developmental transcription factors (TFs) HNF4A and GATA6 roles in this process because of previous findings these are involved in BO. Furthermore, they do snATAC-seq on BO and make the argument that primitive 7 wpc and transitory epithelium in 9 wpc fetal oesophageal foregut epithelium expressed several of the core TFs previously identified in BO and conclude that BO is a reversion to this transitory oesophageal foregut epithelium. They do multiome on fetal oesophageal and gastric tissue and conclude BO is more like fetal oesophageal tissue and not gastric. The fetal nature of BO is an important question, and this study does provide genomic evidence, mostly correlative, that it is the case. The datasets, especially snATAC-seq and multiome, would be useful to the field, despite other fetal oesophageal scRNA-seq datasets being recently published (Yang Y et al. Dev Cell 2025). The main figures mostly focus on the fetal atlas, while almost all BO data/comparisons are Supplemental, which could be reorganized considering the fetal nature of BO is one of the main highlights of manuscript. The manuscript validates a few TFs previously implicated in BO transition, but this could be expanded to discuss additional known TFs (SOX2, GATA4) and potentially identify novel TFs involved in the conversion to and from squamous to columnar. There is some contradictory evidence that confound the conclusion that BO is not a gastric-like conversion and the intestinal-like nature of BO is largely ignored. Using the existing datasets within paper, additional bioinformatic analysis, staining, clarifications, and reinterpretations would improve the manuscript prior to recommendation for publication.

Major Comments:

1. It was unclear where tissues for sequencing was collected and whether putative stomach tissue was included. Sup Fig 1A shows representative images from mid oesophagus and distal oesophagus at different timepoints, which look quite distinct. Figure 1A show mid oesophagus images (duplicated images). It is stated that "lower third of oesophagus was dissected to the gastro-oesophageal junction and upper fundus was dissected from stomach". Was the gastro-oesophageal junction clearly distinguishable in 7 and 9 wks fetal tissue and no part of stomach was included in analysis? Is the thought that the W9w2 Basal cells are from the mid oesophagus and W9c3 Foregut epi is from the distal oesophagus/gastro-oesophageal junction? The transitory epithelium "W9c3 Foregut epithelial" is one of papers main focuses for its BO-like signature (GATA6^{high}TP63^{low}HNF4A^{high}HNF1B^{high}) is consistent with a columnar not squamous identity. This cluster includes a number of traditional gastric mucous cell markers, including TFF1, TFF2, TFF3, MUC5B, MUC5AC, MUC6, CLDN18, etc consistent with differentiated gastric tissue and not unspecified simple columnar or oesophageal epithelium. Likewise, the oesophageal multiome (Sup Fig 12) also had ATP4B FUT8 gastric-like cells. Expanding images in Figure 5I could be helpful. Additional magnifications, additional stains of relevant TFs (HNF4A, GATA6, HNF1B, etc) to help clarify this transitional epithelium would be useful. Knowing where cells with signature of W9c3 are expressed would be important for interpreting data, especially the argument that W9c3 epithelium is not gastric.

2. Considering point 1 above, there is contradictory evidence that the conclusion that BO is a primitive foregut epithelium and not stomach-like. Sup Fig 3D, show BO express KLF5+HNF4A+GATA6+ and claim this signature is not in adult esophagus or stomach tissue so represents fetal primitive and transitory epithelial population, but didn't exclude the intestine. BO is well characterized to have both intestinal and gastric characteristics and Nowicki-Osuch et al. Cancer Discovery 2023 showed that BO metaplasia clusters with adult gastric and/or intestinal cell types and has little to no residual squamous signature. Does the fetal gastric tissue express the key TFs discussed here? The majority of "BO" markers (including Sup Fig 7C) shown are gastric-specific markers (CLDN18 and MUC5AC), but should also look at intestinal markers CDX2, MUC2, etc that wouldn't typically be expressed in squamous/columnar junction during development? HNF4 is a more intestinal-like TF than gastric. Considering the point 1 above, the conclusion that BO or W9c3 does not represent gastric tissue seems incorrect. For oesophagus/stomach multiome (Sup Fig 12-13), why are these not integrated together to more directly compare or at the very least, show

common markers genes between both. Seems strange in Sup Fig 14B that Oesophageal_c3_Foregut_1 expresses more gastric markers (TFF1, TFF2, MUC5AC, MUC5B, CLDN18) than Gastric_c2_foveolar cells or any of the Gastric cell types. Might be worth using adult gastric cells in Sup Figure 14 to confirm gastric_multiome is more similar to those samples as a positive control. The core TFs of BO (Sup Fig 14D) should be included on the same heatmap to more accurately assess levels between tissues. Since ciliated cells not present in BO but are present in 9 wk oesophagus, doesn't this support that BO is a non-oesophageal fate and/or a timepoint (7 wk) prior to emergence of ciliated cells.

3. Not sure why particular markers for dot plot and/or feature plot were included (sometimes most differentially expressed genes not most informative) or discussed in text, but should include additional markers well known in the field. These could include CDH1, SOX2, TP63, cytokeratin of basal and suprabasal layers, IVL, PDPN, FOXF1, PDGFRA, PTCH1, CDH5, etc. Likewise, should have a panel of non-oesophageal/squamous genes, including NKX2-1, GATA4, CLDN18, TFF1, TFF2, TFF3, MUC5AC, MUC6, NKX6-2, PDX1, CDX2, MUC2, etc. Considering the known role of SOX2 and GATA4 in squamous columnar formation/identity these markers should be interrogated further. Cell cycle/mitotic clusters could be identified from the start and could provide earlier insight into RNA velocity data. Additionally, you could use markers identified in BO datasets and look at similar markers in your fetal oesophageal datasets. Other published fetal human oesophageal scRNA-seq atlases available (2nd trimester Yu et al. Cell 2021) and/or the recently published Yang Y et al. Dev Cell 2025, which has a large number of time points could also be useful for additional markers or reanalyzing timepoints to look for similar trends. Yang Y et al should be referenced/discussed.

4. Your group and others have implicated HNF4, HNF1, FOXA, AND GATA4/6 as a core GRN in BO and HNF4A is capable of opening chromatin in squamous to drive acquisition of BO-like signature. Can you use data (fetal or BO) to identify less well characterized GRN involved? Or temporally refine the cascade of TFs shown in Figure 5E.

5. Some of the cluster nomenclature is confusing. A lot of it is based on initial clustering shown in Figure 1C-E. Cluster numbers are based on a specified resolution so numbering is fairly arbitrary. Should just rename as W7 Foregut Epi_1, 2, etc and largely drop cluster numbers. This could be refined following ATAC. You do not need to carry cluster numbers from Figure 1 to all other Figures which was slightly confusing. It might even be mislabeled in some Figures (See Sup Fig 11 W9c2 or W9c3?). Gets even more convoluted when snATAC-seq introduced (See Sup Figure 4A-D, Fig 3-4). Did ATAC lead to the identification of additional clusters or could this be achieved by increasing cluster resolution of snRNA-seq data alone? Furthermore, W9c3 name was changed from "Foregut epi" to "Gastro intestinal" in Figure 5K and Sup Fig 15A contradicting the "not gastric-like" conclusion. Need to clarify annotations throughout with clear stage and cell-type.

Minor Comments:

1. Since manuscript focused on epithelium, maybe whole snRNA-seq atlas data (Fig 1) could all be supplemental (combined with Sup Fig 1 and/or 2). Currently, some sections are fairly redundant. For example, the epithelial specific GATA6High/TP63Low, GATA6Int/TP63Int, GATA6low/TP63high populations are shown and mentioned with whole atlas (Sup Fig 2E) and the epithelial recluster (Fig 2C), when the later would be sufficient. Likewise, when introduce snATAC-seq datasets into analysis for 7 wk (Fig 3, Sup Fig 4-5) and 9 wk (Fig 4, Sup Fig 6-7) they are described separately, but can these, at least text, be combined/condensed. There are not clear conclusions regarding timepoint-specific differences. What is difference between Figure 4C and Sup Figure 7A or duplicate data? Furthermore, the conclusions of entire snATAC-seq section are not definitive, ie the conclusion is just restating epithelium transitions from primitive foregut to ciliated/stratified driven by TFs. Needs to be more impactful. Perhaps a model showing progression of TFs from progenitors to cell types at different time points might be helpful.

2. The whole atlas (Sup Fig 2) and epithelial recluster (Fig 2) show little/no overlap between cell types between timepoints, which could be due to biological/temporal differences or technical due to batch effects. Did you try different integration methods, such as Harmony? Did you integrate entire atlases then subset epithelial data or subset epithelium from individual atlases then integrate? The methods section regarding integration of snRNA-seq datasets is minimal/absent and should be expanded.

3. In mice, SOX2 and GATA4 play a predominate role in specifying KRT7+ transitional epithelium and the squamous-columnar junction is refined throughout midgestation (Sankoda N et al. Nat Commun 2021). It is mentioned there is a "dynamic transition zone develops in lower oesophagus leading to the stomach", but this transitional zone is not clearly discussed when interpreting results. The use of "transitory" to describe 9 wk oesophagus epithelium is easily confused with "transitional" epithelium at junction.

Reviewer 2

SUMMARY OF THE ADVANCE MADE IN THIS PAPER AND ITS POTENTIAL SIGNIFICANCE TO THE FIELD

This manuscript presents a compelling and conceptually provocative hypothesis that Barrett's oesophagus (BO) represents not merely a metaplastic transition but a reversion to an embryonic foregut program, characterized by reactivation of primitive columnar epithelial transcriptional networks. Through single-cell transcriptomic and epigenomic analyses of human fetal oesophageal development, the authors describe a developmental continuum from a programmed squamous epithelium to a simple columnar epithelium, proposing that BO may involve re-engagement of early columnar differentiation programs driven by transcription factors such as HNF4A.

SUGGESTIONS TO AUTHORS

While the data generated is rich and the developmental atlas they present is an important resource for the field, several key limitations in both the interpretation and the experimental framework temper the impact of the conclusions drawn.

1. Lineage Origin and Species Comparisons

The manuscript refers to the transcriptional similarity between BO and early foregut columnar epithelium, implying that the adult oesophageal squamous epithelium can transdifferentiate into BO via reactivation of a fetal program. However, *in vivo* lineage tracing studies in mouse models-particularly the L2-IL-1B and Krt5-CreERT2 lines-have repeatedly demonstrated that BO-like lesions arise not from the oesophageal squamous cells, but from gastric cardia progenitor cells demonstrating that these progenitors migrate proximally into the damaged lower oesophagus during chronic inflammation and give rise to metaplastic epithelium. The manuscript does not address this fundamental conflict with murine lineage evidence, nor does it reconcile how their human fetal transcriptional data can be integrated with these well-established mouse findings.

2. Lack of Functional Validation in Organoid or Tissue Models

The authors convincingly describe transcriptomic and epigenomic profiles across fetal stages, yet fail to test the functional potential of these epithelial states. Specifically, no experimental data from 3D culture models, such as adult or fetal-derived oesophageal or gastric organoids, are presented to confirm that predicted squamous epithelial populations can transition to a columnar phenotype-or vice versa-under defined conditions. Without demonstrating that such reprogramming is possible, the central claim of the manuscript remains speculative. The use of lineage-flexible, inducible systems (i.e. 3D tissue cultures) or co-culture with stromal/immune components would significantly strengthen the hypothesis and bridge developmental and disease contexts.

3. Microenvironmental Influence and Cellular Plasticity

The authors suggest that BO reflects a reactivation of a primitive cell state, but they underappreciate the role of the microenvironment-including inflammation, bile acids, and stromal signaling-in shaping epithelial fate. The question remains: to what extent is the observed transcriptional program cell-intrinsic versus microenvironmentally induced? Could the activation of HNF4A and related networks be driven by extrinsic cues in a chronically inflamed niche, thereby inducing a phenotype that merely resembles fetal epithelium? For instance, studies have shown that bile acids can induce columnar features in oesophageal squamous cells *in vitro*, but whether this is due to reactivation of a fetal program or an adaptive stress response is unclear.

4. Interpreting Metaplasia as Protection or Progression

Another important point unaddressed is whether metaplasia represents a protective, adaptive response to chronic injury rather than a pre-malignant or embryonic reversion. The notion that

progenitor cells-either from the oesophagus or gastric cardia-may differentiate into a columnar lineage under injury conditions could be a form of barrier reinforcement rather than a direct path to dysplasia and cancer. This distinction is crucial, especially given that not all BO progresses to EAC. A more nuanced discussion of how microenvironmental context, genetic alterations (e.g., TP53 or CDKN2A loss), and tissue stress might determine the trajectory toward cancer is needed.

5. The Role of Transcriptional Regulation vs Environmental Cues

Finally, while the enrichment of BO-associated GWAS SNPs in fetal epithelial regulatory regions is intriguing, it remains unclear whether these regions are causally implicated in disease or simply correlated with shared chromatin states. Given that chromatin accessibility is highly sensitive to environmental stimuli, the manuscript should discuss whether transcription factor profiles such as those regulated by HNF4A are truly reactivated in a cell-intrinsic manner or defined by epigenetic remodeling driven by the microenvironment. Moreover, can the adult oesophageal epithelium, in the presence of an embryonic-like microenvironment, be coaxed into reactivating such programs?

In summary, while the transcriptomic and epigenomic mapping of fetal oesophageal development is a valuable contribution, the central claim-that BO represents a reactivation of primitive fetal states-requires more rigorous functional validation and reconciliation with existing lineage-tracing data. A more integrative framework that considers the contribution of adult progenitor plasticity, environmental cues, and inter-tissue boundaries (e.g., gastric vs oesophageal origin) would render this study more impactful and relevant for the understanding of BO pathogenesis. As it stands, the study opens an interesting conceptual avenue but lacks the experimental and translational depth needed to substantiate its core hypothesis.

Reviewer 3

SUMMARY OF THE ADVANCE MADE IN THIS PAPER AND ITS POTENTIAL SIGNIFICANCE TO THE FIELD

This manuscript from Baker et al., seeks to establish transcriptome-wide understanding of esophageal epithelial development and its relationship to the pathological condition, Barrett's esophagus (BO), in which the esophageal epithelium transforms from a squamous to more columnar histology, and can be a precancerous hallmark. While several hypotheses on the cellular basis for these changes have been made, detailed molecular analysis to address these hypotheses has been lacking. Here, by performing single nuclei RNA-seq and ATAC seq on progressive stages of human development, the authors are able to delineate likely cell fate transitions during development, showing transitions through primitive foregut to transitional, ciliated and basal stratified epithelial cell types. Comparison of snATAC-seq datasets with RNA-seq gene expression using bioinformatic approaches characterize some of the key transcription factor regulators (HNF4A, GRHL1, TP63) and associated chromatin changes underlying cell fate transitions at these stages of development. Comparison of these datasets to previously published single cell RNA-seq and newly-generated snATAC-seq datasets from patients was able to map cell types found in BO to developmental populations and enabled the interrogation of previously implicated BO regulatory TFs (GATA4/6, HNF1A, HNF4A) in the differentiation of BO cell types; these transcription factors were also found to be expressed in correlation with cell fate transitions in development, further supporting the concept that BO could represent a form of reversion to transient epithelial populations present in development. Comparison of BO with epithelial stomach populations further reinforced that BO more transcriptionally resembles developmental esophageal cell types, rather than those from the stomach. As this work provides the first snRNA-seq and snATAC seq atlas of human esophageal development, and novel BO snATAC-seq datasets, this manuscript provides invaluable new data for the study of esophageal development and BO. Furthermore, the use of sophisticated bioinformatic tools enable insights into the basis of BO pathogenesis and indicate that it likely arises as a form of developmental reversion to a transitional cell state. The manuscript is very well-written and figures are mostly clear. This will be a valuable resource and the findings will be of significant interest to developmental biology and gastroenterology communities. A few minor points might be considered for whether addressing them might improve this strong manuscript.

SUGGESTIONS TO AUTHORS

- * As I understand it, combined datasets in single UMAPs in Supplementary Fig. 2, and re-clustered UMAP in Fig 2 do not include integration across stage correction, and indeed cluster analysis seems to predominantly identify stage differences. It might be informative to additionally perform integration across stages to minimize stage effects in favor of cell-type specific information.
- * Text on page 5 describes a CFAP73 high ciliated and low population and points to Fig. 2B (where the only similar marker is "CFAP45", and Fig. 2C and Supp. Fig. 3D "CFAP43" ciliated cells are highlighted in the UMAP. I suspect these are typos and all should refer to the same marker of ciliated cells.
- * It might be beneficial to discuss the extent to which functional experiments in mouse validate the developmental transcription factor expression/motif accessibility trajectories described in Figure 3. There may be other opportunities to further reference mouse functional data as it relates to the findings here as well.
- * While not particularly problematic, the inclusion of GWAS SNP analysis in this report is considered somewhat speculative and does not greatly advance the main points of this study; that 4/27 of the GWAS SNPs map to cis-regulatory elements relevant to esophageal epithelial development does not lend additional support to the origin of BO, especially as other GWAS SNPs may overlap well with cis-regulatory elements in gastric epithelium or development.

First revision

Author response to reviewers' comments

Reviewer 1:

The referee concludes that *"we provide genomic evidence for the fetal nature of BO"* and that *"we provide datasets that would be useful to the field, despite other fetal oesophageal scRNA-seq datasets being recently published"*. He/she recommends *"using the existing datasets within paper, additional bioinformatic analysis, staining, clarifications, and reinterpretations would improve the manuscript prior to recommendation for publication"*.

Response: We appreciate the overall positive evaluation of our work and have followed all of the referees' recommendations to improve our work as outlined below.

The following issues were raised in their overview:

He/she suggests *"The main figures mostly focus on the fetal atlas, while almost all BO data/comparisons are Supplemental, which could be reorganized considering the fetal nature of BO is one of the main highlights of manuscript"*.

Response: This is a positive suggestion and we have now moved Supplementary Fig.S9 which reports new snATAC-seq data on BO to the main manuscript in recognition of this (now Fig. 5).

The manuscript validates a few TFs previously implicated in BO transition, but this could be expanded to discuss additional known TFs (SOX2, GATA4) and potentially identify novel TFs involved in the conversion to and from squamous to columnar.

Response: We have discussed these two TFs in response to point 3 below. We are reluctant to speculate further on potential squamous-columnar transitions as this may not be the route for developing BO. We have expanded our discussion though of the implications of our study regarding transition from a cell of origin, with our finding demonstrating the likely endpoint, rather than the start point of the process. Further extensive work will be required to unpick the molecular mechanisms involved.

There is some contradictory evidence that confound the conclusion that BO is not a gastric-like conversion and the intestinal-like nature of BO is largely ignored.

Response: We report here the molecular similarities between BO and early developmental foregut. This is not contradictory to the prevailing view that BO is intestinal-like and not gastric-like. Indeed, the latest paper on the subject implies that adult gastric foveolar cells are converted into BO, and hence acquire a broader more intestinal-like property. We have ensured that the text makes it clear that our data do not rule in one cell type rather than another as the cell of origin but that whatever cell it is, then reverts back to something that is developmental-like in nature. The epigenetic barriers to this will be different in different cell types, so there may be several different cells of origin with some more common than others due to the intrinsic epigenetic barriers.

Major issues:

1. *It was unclear where tissues for sequencing was collected and whether putative stomach tissue was included. Sup Fig 1A shows representative images from mid oesophagus and distal oesophagus at different timepoints, which look quite distinct. Figure 1A show mid oesophagus images (duplicated images). It is stated that "lower third of oesophagus was dissected to the gastro-oesophageal junction and upper fundus was dissected from stomach". Was the gastro-oesophageal junction clearly distinguishable in 7 and 9 wks fetal tissue and no part of stomach was included in analysis?*

Response: Our dissections were performed as stated in the text and highlighted by the referee. We have rewritten the section for clarity and it now reads:

"In brief, for oesophageal samples, the lower third of the oesophagus was dissected to the gastro-oesophageal junction, and for gastric samples, the upper fundus was dissected from the stomach."

We are confident that our initial dissections used for the paired RNA-seq and ATAC-seq samples, did not contain any stomach material. This is also illustrated, in our multiome analysis of gastric and oesophageal samples where the epithelial cell types generally did not overlap between the stomach and oesophagus samples (see new Supplementary Fig. 13). However, we did identify a small cell population in our oesophageal multiome sample that we did not find from our paired RNA- and ATAC-seq samples which showed overlap with gastric cell types, suggesting potential small amounts of contamination (discussed further below).

Is the thought that the W9w2 Basal cells are from the mid oesophagus and W9c3 Foregut epi is from the distal oesophagus/gastro-oesophageal junction? The transitory epithelium "W9c3 Foregut epithelial" is one of papers main focuses for its BO-like signature (GATA6^{high}TP63^{low}HNF4A^{high}HNF1B^{high}) is consistent with a columnar not squamous identity. This cluster includes a number of traditional gastric mucous cell markers, including TFF1, TFF2, TFF3, MUC5B, MUC5AC, MUC6, CLDN1B, etc consistent with differentiated gastric tissue and not unspecified simple columnar or oesophageal epithelium. Likewise, the oesophageal multiome (Sup Fig 12) also had ATP4B FUTB gastric-like cells. Expanding images in Figure 5I could be helpful.

Response: As indicated above, the multiome contains an additional cluster not seen in our original datasets and this small cluster is the one that expresses *ATP4B* and *FUTB* and clusters with gastric epithelial cells (see Fig. S12G). We have now drawn further attention to this cluster in the manuscript with the possibility that this is contamination from the stomach. We have added additional UMAPs, generated from combining the multiome data from the oesophageal and gastric epithelial populations, that clearly demonstrate how this sub-population clusters with the gastric cell populations rather than the oesophageal populations (which remain distinct) (Supplementary Fig. S13A-C) and lend weight to this hypothesis. Importantly though we still identify the three major epithelial populations in the oesophagus that do not overlap with gastric populations (See also original Supplementary Fig.13G). We are therefore confident in their designation as oesophageal epithelial populations.

To further reinforce this point, we have added additional UMAPs with oesophageal-specific, shared and gastric-specific markers on them, to demonstrate the uniqueness of our primitive foregut population in the oesophagus (Supplementary Fig. 13D-F). We have also added additional IHC showing MUC5AC staining across several upper GI locations which effectively extends the field of view of Fig. 5I (new Supplementary Fig.13G).

Additional magnifications, additional stains of relevant TFs (HNF4A, GATA6, HNF1B, etc) to help clarify this transitional epithelium would be useful. Knowing where cells with signature of W9c3 are expressed would be important for interpreting data, especially the argument that W9c3 epithelium is not gastric.

Response: We attempted to stain for the transcription factors but could not obtain good signal, potentially due to the quality of antibodies. However, instead, we added extra IHC figures for MUC5AC staining. This is a marker of the oesophageal primitive foregut cells, and we can see clear expression in the lower oesophagus in 9 week embryos (new Supplementary Fig. S13G). Thus, in addition to their known expression in the stomach and GOJ, we detect additional expression in oesophageal cells, in keeping with our findings from dissociated oesophageal cells in the snRNA-seq. This provides definitive proof that the MUC5AC expressing W9c3 primitive foregut epithelial cells we uncovered are located in the oesophagus and not in the stomach.

2. Considering point 1 above, there is contradictory evidence that the conclusion that BO is a primitive foregut epithelium and not stomach-like. Sup Fig 3D, show BO express KLF5+HNF4A+GATA6+ and claim this signature is not in adult esophagus or stomach tissue so represents fetal primitive and transitory epithelial population, but didn't exclude the intestine. BO is well characterized to have both intestinal and gastric characteristics and Nowicki-Osuch et al. Cancer Discovery 2023 showed that BO metaplasia clusters with adult gastric and/or intestinal cell types and has little to no residual squamous signature. Does the fetal gastric tissue express the key TFs discussed here? The majority of "BO" markers (including Sup Fig 7C) shown are gastric-specific markers (CLDN1B and MUC5AC), but should also look at intestinal markers CDX2, MUC2, etc that wouldn't typically be expressed in squamous/columnar junction during development?

Response: We have already analysed the expression of the key TFs in gastric tissue and the data were shown in original supplementary Fig. S14D-E. We have now shown this in a side-by-side comparison with oesophageal epithelial clusters (extra panel in new supplementary Fig. 14D). Note that the TFs are also expressed in the stomach, albeit to different extents in different cell types, in contrast to their co-expression in the foregut epithelial cell population (see Supplementary Fig 14D). We have added further discussion of the implications of this for linking to gastric epithelial cells as the potential cell of origin in the adult.

HNF4 is a more intestinal-like TF than gastric. Considering the point 1 above, the conclusion that BO or W9c3 does not represent gastric tissue seems incorrect. For oesophagus/stomach multiome (Sup Fig 12-13), why are these not integrated together to more directly compare or at the very least, show common markers genes between both. Seems strange in Sup Fig 14B that Oesophageal_c3_Foregut_1 expresses more gastric markers (TFF1, TFF2, MUC5AC, MUC5B, CLDN1B) than Gastric_c2_foveolar cells or any of the Gastric cell types.

Response: We have taken this suggestion and now integrated all the epithelial cells from the multiome data together (New supplementary Fig. 13A-C). We have added further UMAPs from the combined multiome data with expression of shared and distinct marker genes superimposed (new supplementary Fig. 13D-F) which reinforces the conclusions from Supplementary Fig.14B. There are shared markers like *CLDN1B*, unique oesophageal markers like *CPNEB*, and unique gastric markers like *ATP4A*. This clearly demonstrates that although some markers are shared, the oesophageal and gastric populations show unique molecular characteristics in terms of their gene expression profiles. This is now discussed further in the text.

Might be worth using adult gastric cells in Sup Figure 14 to confirm gastric_multiome is more similar to those samples as a positive control.

Response: We compared the epithelial cell clusters from our oesophageal and gastric multiomes

with the adult gastric single cell data, but no clear pattern was observed. Indeed, only low-level matches were observed between the developmental and adult gastric cell populations, likely representing the very different states they adopt during development and in the adult. This is coupled with technical differences (multiome versus scRNA-seq, snRNA-seq versus scRNA-seq, cell dissociation versus nuclear isolation etc) between the different studies. As we are not investigating gastric development in our study, this topic is best returned to at a later date with more controlled sample acquisition, and later developmental timepoints to address this specific question of how stomach development proceeds.

The core TFs of BO (Sup Fig 14D) should be included on the same heatmap to more accurately assess levels between tissues.

Response: We have added an additional panel to Sup Fig 14D (previously 14D) that directly compares across tissues so that the relative levels can be assessed. We retained the original panels to demonstrate intra-tissue differences in expression. The story remains unchanged *Since ciliated cells not present in BO but are present in 9 wk oesophagus, doesn't this support that BO is a non-esophageal fate and/or a timepoint (7 wk) prior to emergence of ciliated cells.*

Response: While the Barrett's samples analysed did not contain ciliated cells, other studies have begun to uncover ciliated epithelial cells in BO cases. These are thought to be located more towards the periphery of the lesion, hence why sampling likely often misses these. A reference to a recent pre-print (Gier et al., 2023; doi: 10.1101/2023.01.26.525564) and discussion of this has now been added to the paper.

3. Not sure why particular markers for dot plot and/or feature plot were included (sometimes most differentially expressed genes not most informative) or discussed in text, but should include additional markers well known in the field. These could include CDH1, SOX2, TP63, cytokeratin of basal and suprabasal layers, IVL, PDPN, FOXF1, PDGFRA, PTCH1, CDH5, etc.

Response: We checked all these markers and few are expressed at high levels in the epithelial layers in our datasets and are uninformative. *TP63* was already included in original Fig.2C and discussed in the text. *CDH1* is expressed in all epithelial cell types so we have replaced *KLF5* with this more well studied epithelial marker in Fig.2C as a pan epithelial marker. We have however included the keratins to delineate the basal and suprabasal layers of the stratified epithelial layer (new Fig. 2E and F) to provide a more fine-grained view of these cell types as development progresses and enable cross comparison to the data in Yang et al., 2025.

Likewise, should have a panel of non-esophageal/squamous genes, including NKX2-1, GATA4, CLDN1B, TFF1, TFF2, TFF3, MUC5AC, MUC6, NKX6-2, PDX1, CDX2, MUC2, etc. Considering the known role of SOX2 and GATA4 in squamous columnar formation/identity these markers should be interrogated further.

Response: We have also added a panel of these genes which are characteristic of different epithelial types found in different parts of the gut (New supplementary Fig. 3D). We also added a new figure (new supplementary Fig. S13H) to illustrate the demarcation of the squamous layer from the stomach in the transitional stage using the markers requested ie *GATA4* for gastric columnar and *SOX2* for squamous oesophageal. These are expressed as expected from previous work in mice, and this is now referenced and discussed in the text.

Cell cycle/mitotic clusters could be identified from the start and could provide earlier insight into RNA velocity data. Additionally, you could use markers identified in BO datasets and look at similar markers in your fetal oesophageal datasets. Other published fetal human oesophageal scRNA-seq atlases available (2nd trimester Yu et al. Cell 2021) and/or the recently published, which has a large number of time points could also be useful for additional markers or reanalyzing timepoints to look for similar trends. Yang Y et al should be referenced/discussed.

Response: We have added discussion of the very recent paper Yang Y et al. *Dev Cell* 2025 which substantiates the major findings from our study. One key difference is that they only report ciliated and cells constituting the squamous stratified epithelial layer, which we attribute mainly due to their later sampling and/or technical differences in sample acquisition and processing. We

have also added discussion of data from Yu et al. *Cell*, 2021, although this is limited to 14 weeks onwards, corresponding to the last latest sample in our dataset. We already have included well characterised Barrett's markers in our study and checked these in our developmental datasets (Supplementary Fig. S14B).

The referee makes a good point about the cycling cells which we did not take into account when doing the RNA velocity analysis. We previously illustrated the presence of a cycling population in Supplementary Fig. 4C-E in the 7 week epithelial cells, and have now removed this and redone the velocity analysis (new Supplementary Fig. 4H). This does not greatly alter the overall velocity directionality in the UMAP, and we have revised the manuscript text accordingly to include this new analysis. We also added cell cycle analysis for the 9 week epithelial cells (new Supplementary Fig. 6E) but few cycling cells are present by this stage, and they do not affect the overall directionality indicated by the velocity analysis.

4. Your group and others have implicated HNF4, HNF1, FOXA, AND GATA4/6 as a core GRN in BO and HNF4A is capable of opening chromatin in squamous to drive acquisition of BO-like signature. Can you use data (fetal or BO) to identify less well characterized GRN involved? Or temporally refine the cascade of TFs shown in Figure 5E.

Response: Figure 5E (now 6E) already implies a temporal order of action of these TFs. We have identified numerous new TFs beyond the "BO network" that are implicated in oesophageal epithelial development (Fig.3C&F; Supplementary Fig. 7C) and we have now added important discussion of the potential to discover new regulators in the revised text. While tempting to do, we do not think it is worthwhile to computationally attempt to predict drivers of BO conversion. First because the cell of origin might not be (and is unlikely to be) adult squamous cells, and secondly as functional validation would be needed which is beyond the scope of the current paper.

5. Some of the cluster nomenclature is confusing. A lot of it is based on initial clustering shown in Figure 1C-E. Cluster numbers are based on a specified resolution so numbering is fairly arbitrary. Should just rename as W7 Foregut Epi_1, 2, etc and largely drop cluster numbers. This could be refined following ATAC. You do not need to carry cluster numbers from Figure 1 to all other Figures which was slightly confusing. It might even be mislabeled in some Figures (See Sup Fig 11 W9c2 or W9c3?). Gets even more convoluted when snATAC-seq introduced (See Sup Figure 4A-D, Fig 3-4).

Response: The referee makes a good point and we have simplified things in several places. However, for ease of cross referencing and labelling, we have generally retained the cluster numbers in the figures but now refer to the cluster names in the text rather than just numbers to make things clearer. We have also updated Fig.3A, Fig.4A&E and supplementary Fig.4A, Fig.6A&B and removed RNA-seq cluster numbers and just retained the names to make this clearer when discussing new ATAC-derived clusters (with the new cluster numbers retained for ease of reference). We have corrected the labelling in Sup Fig 11.

Did ATAC lead to the identification of additional clusters or could this be achieved by increasing cluster resolution of snRNA-seq data alone?

Response: As indicated in the paper, the integration of our ATAC- and RNA-seq data allowed new "transitory" cell populations to be uncovered in our 9 week sample. This led us to recluster the RNA-seq data (see supplementary Fig 6C already in original manuscript) which helped inform our RNA trajectory analysis. We would not have done this reclustering unless prompted by the ATAC data, demonstrating the utility of incorporating two different modalities. Prompted by the findings from our joint modality integration we did however recluster the RNA-seq data (already in previous version of paper; supplementary Fig.6C) The findings from that agreed with what we discovered from the joint modality. We have now added new UMAPs in Supplementary Fig. 6D with marker TFs for each of the states identified in Fig.4A.

Furthermore, W9c3 name was changed from "Foregut epi" to "Gastro intestinal" in Figure 5K and Sup Fig 15A contradicting the "not gastric-like" conclusion. Need to clarify annotations throughout with clear stage and cell-type.

Response: We apologise for this oversight and have corrected the nomenclature in this figure which was initially brought in to reflect the similarities to Barrett's and then renamed to avoid the confusion that you rightly point out.

Minor concerns:

1. *Since manuscript focused on epithelium, maybe whole snRNA-seq atlas data (Fig 1) could all be supplemental (combined with Sup Fig 1 and/or 2). Currently, some sections are fairly redundant. For example, the epithelial specific GATA6High/TP63Low, GATA6Int/TP63Int, GATA6low/TP63high populations are shown and mentioned with whole atlas (Sup Fig 2E) and the epithelial recluster (Fig 2C), when the later would be sufficient.*

Response: We would like to retain the current structure as the whole atlas is an important deliverable from this project. We agree these parts are redundant but think this reinforces the message and helps define defining features of cell clusters at an early stage.

Likewise, when introduce snATAC-seq datasets into analysis for 7 wk (Fig 3, Sup Fig 4-5) and 9 wk (Fig 4, Sup Fig 6-7) they are described separately, but can these, at least text, be combined/condensed. There are not clear conclusions regarding timepoint-specific differences.

Response: Describing each stage sequentially through developmental time is our preferred option so we would like to retain the current structure (where things are combined in the same section. We have however now emphasised the differences between timepoints in the results section, and the model in new Fig. 4H helps summarise the findings.

What is difference between Figure 4C and Sup Figure 7A or duplicate data?

Response: These are the same data. The data in supplemental contains additional TF names. Adding these to the main figure in sufficient size and resolution would increase the figure size so we think it best to stay with this presentation style which provides a clearer message in the main figures.

Furthermore, the conclusions of entire snATAC-seq section are not definitive, ie the conclusion is just restating epithelium transitions from primitive foregut to ciliated/stratified driven by TFs. Needs to be more impactful. Perhaps a model showing progression of TFs from progenitors to cell types at different time points might be helpful.

Response: This is a useful suggestion and we now added an informative model (new Fig.4H) and included this in the concluding statement.

2. *The whole atlas (Sup Fig 2) and epithelial recluster (Fig 2) show little/no overlap between cell types between timepoints, which could be due to biological/temporal differences or technical due to batch effects. Did you try different integration methods, such as Harmony? Did you integrate entire atlases then subset epithelial data or subset epithelium from individual atlases then integrate? The methods section regarding integration of snRNA-seq datasets is minimal/absent and should be expanded.*

Response: Given that these are different developmental timepoints and that the tissue is undergoing dramatic remodelling, it is not surprising that the epithelial layer is changing in terms of molecular characteristics of cells. However, in Sup Fig 2A&B it is clear that the non- epithelial cells do cluster together fairly well whereas there are clear distinctions between the epithelial cells. This is also apparent in the markers of the clusters which largely do not overlap ie there are clearly distinct epithelial clusters for squamous, foregut and ciliated cells. These also change over time, and it is illustrated by the two populations of ciliated cells at week 15 but only one at week 9. We have updated the methods section and provided more detail about how the snRNA-seq datasets were integrated. Importantly we also tried integrating the datasets using Harmony and the distinctions between epithelial populations and different timepoints remained.

3. In mice, SOX2 and GATA4 play a predominate role in specifying KRT7+ transitional epithelium and the squamous-columnar junction is refined throughout midgestation (Sankoda N et al. Nat Commun 2021). It is mentioned there is a "dynamic transition zone develops in lower oesophagus leading to the stomach", but this transitional zone is not clearly discussed when interpreting results. The use of "transitory" to describe 9 wk oesophagus epithelium is easily confused with "transitional" epithelium at junction.

Response: We can appreciate why this might be confusing. However, the study by Sankoda details the expression of SOX2 (distal) and GATA4 (proximal) in the stomach rather than the oesophagus and how these regulate SCJ formation. Here we are discussing events close to the SCJ and have reworded the text accordingly and refer to the SCJ as a boundary and the epithelium at 9 weeks as "transitory" as it is neither squamous nor columnar but is of a mixed nature. We also examined GATA4 expression and confirmed this is largely confined to the gastric epithelial populations whereas SOX2 and KRT7 are more expressed in the oesophageal squamous epithelial cells (new data in Supplementary Fig. 13H) and is discussed in the text with reference to the mouse data.

Reviewer 2:

The referee concludes that "*This manuscript presents a compelling and conceptually provocative hypothesis that Barrett's oesophagus (BO) represents not merely a metaplastic transition but a reversion to an embryonic foregut program, characterized by reactivation of primitive columnar epithelial transcriptional networks*".

Response: We appreciate the overall positive evaluation of our work and have responded to the issues raised as outlined below.

The following issues were raised:

While the data generated is rich and the developmental atlas they present is an important resource for the field, several key limitations in both the interpretation and the experimental framework temper the impact of the conclusions drawn.

1. *Lineage Origin and Species Comparisons: The manuscript refers to the transcriptional similarity between BO and early foregut columnar epithelium, implying that the adult oesophageal squamous epithelium can transdifferentiate into BO via reactivation of a fetal program. However, in vivo lineage tracing studies in mouse models-particularly the L2-IL-1f3 and Krt5-CreERT2 lines-have repeatedly demonstrated that BO-like lesions arise not from the oesophageal squamous cells, but from gastric cardia progenitor cells demonstrating that these progenitors migrate proximally into the damaged lower oesophagus during chronic inflammation and give rise to metaplastic epithelium. The manuscript does not address this fundamental conflict with murine lineage evidence, nor does it reconcile how their human fetal transcriptional data can be integrated with these well-established mouse findings.*

Response: We have been careful to not "imply that the adult oesophageal squamous epithelium can transdifferentiate into BO via reactivation of a fetal program" as the referee states but have instead suggested this as one of several possibilities, each supported by differing amounts of published evidence. This was in the discussion of the original paper and we have now bolstered this further to avoid misinterpretations. In this more nuanced discussion, we emphasise that the gastric source of BO in the adult is entirely possible and is not inconsistent with our data. The key is that an adult cell type(s)/state(s) reverts to an earlier developmental timepoint, with each one providing differing epigenetic barriers that need to be overcome. Some are more susceptible than others of course but theoretically any cell might revert given sufficient selective pressure and molecular changes.

2. *Lack of Functional Validation in Organoid or Tissue Models The authors convincingly describe transcriptomic and epigenomic profiles across fetal stages, yet fail to test the functional potential of these epithelial states. Specifically, no experimental data from 3D culture models, such as adult or fetal-derived oesophageal or gastric organoids, are presented to confirm that predicted squamous epithelial populations can transition to a columnar phenotype-*

or vice versa-under defined conditions. Without demonstrating that such reprogramming is possible, the central claim of the manuscript remains speculative. The use of lineage-flexible, inducible systems (i.e. 3D tissue cultures) or co-culture with stromal/immune components would significantly strengthen the hypothesis and bridge developmental and disease contexts.

Response: We appreciate the referee's concerns here, but this level of additional work is outside the scope of this already extensive manuscript. We have recently sought funding to do these experiments but the timeframe was three years to do adequate functional testing and would represent a new paper that provides detailed mechanistic insights. It is important to acknowledge the limitations of the work and have now done so in the discussion. Nevertheless, we remain convinced that we have provided a valuable resource which makes testable predictions supported by detailed computational analysis.

3. Microenvironmental Influence and Cellular Plasticity: The authors suggest that BO reflects a reactivation of a primitive cell state, but they underappreciate the role of the microenvironment-including inflammation, bile acids, and stromal signaling-in shaping epithelial fate. The question remains: to what extent is the observed transcriptional program cell-intrinsic versus microenvironmentally induced? Could the activation of HNF4A and related networks be driven by extrinsic cues in a chronically inflamed niche, thereby inducing a phenotype that merely resembles fetal epithelium? For instance, studies have shown that bile acids can induce columnar features in oesophageal squamous cells in vitro, but whether this is due to reactivation of a fetal program or an adaptive stress response is unclear.

Response: The referee again raises an important research question but this is also outside the scope of the current study. Linked to this, the cell of origin in the adult still remains unclear at this stage (which we have made clear in the manuscript), and it might well be gastric cells. What we have shown is that reversion occurs to an early fetal stage. How this occurs is a completely different research question and again will be part of future studies. We have again included this important area for future investigation in our discussion, illustrating the importance of our data for informing such studies.

4. Interpreting Metaplasia as Protection or Progression: Another important point unaddressed is whether metaplasia represents a protective, adaptive response to chronic injury rather than a pre-malignant or embryonic reversion. The notion that progenitor cells-either from the oesophagus or gastric cardia-may differentiate into a columnar lineage under injury conditions could be a form of barrier reinforcement rather than a direct path to dysplasia and cancer. This distinction is crucial, especially given that not all BO progresses to EAC. A more nuanced discussion of how microenvironmental context, genetic alterations (e.g., TP53 or CDKN2A loss), and tissue stress might determine the trajectory toward cancer is needed.

Response: This is another good point, and we have added additional discussion of this in the paper. Again, it is important to point out that whatever the mechanism for metaplasia, the endpoint is what we observe ie similar to the early developing oesophagus. This is likely due to the limited epigenetic states that cells at the SCJ can sample for repurposing. This state is presumably formed as a protective response as the referee states, and there is an selective pressure to attain and maintain this state due to ongoing chronic injury.

5. The Role of Transcriptional Regulation vs Environmental Cues: Finally, while the enrichment of BO-associated GWAS SNPs in fetal epithelial regulatory regions is intriguing, it remains unclear whether these regions are causally implicated in disease or simply correlated with shared chromatin states. Given that chromatin accessibility is highly sensitive to environmental stimuli, the manuscript should discuss whether transcription factor profiles such as those regulated by HNF4A are truly reactivated in a cell-intrinsic manner or defined by epigenetic remodeling driven by the microenvironment.

Response: The referee is correct in that we cannot say that SNPs in embryonic/fetal enhancers give rise to the disease (and we have not said this). However, the fact that these chromatin regions are open and potentially active during development and not in adult tissue, makes this a distinct possibility. The microenvironment and stimuli within that is very likely to trigger these epigenetic changes, likely in a stepwise manner and maybe through HNF4A, and we have now

discussed this possibility more in the manuscript. We also removed the mention of GWAS signals from the abstract to de-emphasise this conclusion.

Moreover, can the adult oesophageal epithelium, in the presence of an embryonic-like microenvironment, be coaxed into reactivating such programs?

Response: This is an interesting question but well beyond the scope of the current study.

In summary, while the transcriptomic and epigenomic mapping of fetal oesophageal development is a valuable contribution, the central claim-that BO represents a reactivation of primitive fetal states-requires more rigorous functional validation and reconciliation with existing lineage-tracing data. A more integrative framework that considers the contribution of adult progenitor plasticity, environmental cues, and inter-tissue boundaries (e.g., gastric vs oesophageal origin) would render this study more impactful and relevant for the understanding of BO pathogenesis. As it stands, the study opens an interesting conceptual avenue but lacks the experimental and translational depth needed to substantiate its core hypothesis.

Response: Our data clearly demonstrate that Barrett's resembles an embryonic/fetal state, both in its core transcriptional regulatory networks and the expression of well characterised marker genes. We agree that we have not shown how adult cells transition to this state but this is beyond the scope of the current study. We have however discussed the caveats and integrated in more discussion of existing knowledge in this area.

Reviewer 3:

This reviewer is very positive about our paper and concludes: "*As this work provides the first snRNA-seq and snATAC seq atlas of human esophageal development, and novel BO snATAC-seq datasets, this manuscript provides invaluable new data for the study of esophageal development and BO. Furthermore, the use of sophisticated bioinformatic tools enable insights into the basis of BO pathogenesis and indicate that it likely arises as a form of developmental reversion to a transitional cell state. The manuscript is very well-written and figures are mostly clear. This will be a valuable resource and the findings will be of significant interest to developmental biology and gastroenterology communities. A few minor points might be considered for whether addressing them might improve this strong manuscript.*"

Minor issues raised:

1. *As I understand it, combined datasets in single UMAPs in Supplementary Fig. 2, and re-clustered UMAP in Fig 2 do not include integration across stage correction, and indeed cluster analysis seems to predominantly identify stage differences. It might be informative to additionally perform integration across stages to minimize stage effects in favor of cell-type specific information.*

Response: We responded to a similar comment raised by the first reviewer, and the batch corrections we tried (here effectively the timepoints are the batches) did little to change the clustering. Given that these are different developmental timepoints and that the tissue is undergoing dramatic remodelling, it is not surprising that the epithelial layer is changing in terms of molecular characteristics of cells. However, in Sup Fig 2A&B it is clear that the non-epithelial cells do cluster together fairly well whereas there are clear distinctions between the epithelial cells. This is also apparent in the markers of the clusters which largely do not overlap ie there are clearly distinct epithelial clusters for squamous, foregut and ciliated cells. These also change over time, and it is illustrated by the two populations of ciliated cells at week 15 but only one at week 9.

2. *Text on page 5 describes a CFAP73 high ciliated and low population and points to Fig. 2B (where the only similar marker is "CFAP45", and Fig. 2C and Supp. Fig. 3D "CFAP43" ciliated cells are highlighted in the UMAP. I suspect these are typos and all should refer to the same marker of ciliated cells.*

Response: A lot of genes starting with "CFAP" pre-fix are specifically expressed in ciliated cells, and our initial figures selected different examples. However, these should have been consolidated to focus on CFAP73 for clarity and the wrong figures were submitted. This error has now been corrected on the UMAPs. We retained CFAP45 and CFAP54 in Fig.2B as this represents an unbiased presentation of the top marker genes for each cell cluster. We have also rewritten the text, to reflect which facts come from which subpart of the figure as it was previously unclear.

3. It might be beneficial to discuss the extent to which functional experiments in mouse validate the developmental transcription factor expression/motif accessibility trajectories described in Figure 3. There may be other opportunities to further reference mouse functional data as it relates to the findings here as well.

Response: We have added further discussion about how our work relates to what is known about transcription factor networks from mice. To date there is no evidence for a role of the core HNF4A-GATA-HNF1-FOXA network in oesophageal development, further emphasising the importance of our study where we focus on human development which diverges from that seen in mice. However, GATA6 has been linked to HNF4A during early endoderm differentiation in mice and we now make reference to this in the paper. In response to referee 1, we now have added data on GATA4 and SOX2 expression in the human oesophagus as this complements what has been found in mice where these TFs define the GOJ (new supplementary 13H).

4. While not particularly problematic, the inclusion of GWAS SNP analysis in this report is considered somewhat speculative and does not greatly advance the main points of this study; that 4/27 of the GWAS SNPs map to cis-regulatory elements relevant to esophageal epithelial development does not lend additional support to the origin of BO, especially as other GWAS SNPs may overlap well with cis-regulatory elements in gastric epithelium or development.

Response: The referee raises a good point and it is possible that these regions may be active in other developmental tissues as well. However, the fact that these are accessible in oesophageal epithelial cells during development and not in the adult oesophagus is suggestive of a functional link and raises the possibility that defective gene expression during early development may lead to disease in later life. We have been careful not to conclude any further than this possibility and further functional testing of the consequences of the SNPs for enhancer activity, gene expression effects and the role of the gene products in potential oesophageal epithelial cell phenotypes and functionality will be needed to definitively prove causative effects. We have therefore elected to retain this data in the paper as it provides insights into how these SNPs might function through the non-coding genome. We have however removed mention of GWAS matches from the abstract to de-emphasise this conclusion.

Second decision letter

MS ID#: dev.204735R1

MS Title: The metaplastic precursor to oesophageal adenocarcinoma represents reversion to a transient developmental-like epithelial cell state

Authors: Syed Baker; Aoibheann Mullan; Rachel Jennings; Karen Piper-Hanley; Yeng Ang; Claire Palles; Neil Hanley; Andrew David Sharrocks
Article Type: Research Article

Dear Dr Sharrocks,

I have now received all the referees reports on the above manuscript, and have reached a decision. The referees' comments are appended below, or you can access them online: please go to .

The overall evaluation is positive and we would like to publish a revised manuscript in Development, provided that the referees' comments can be satisfactorily addressed. In particular,

reviewer 1 noticed some missing labels, and had a few textual suggestion to clarify the figures, which I agree should be added. Please attend to all of the reviewers' comments in your revised manuscript and detail them in your point-by-point response. If you do not agree with any of their criticisms or suggestions explain clearly why this is so. If it would be helpful, you are welcome to contact us to discuss your revision in greater detail. Please send us a point-by-point response indicating your plans for addressing the referees' comments, and we will look over this and provide further guidance.

Reviewer 1

Baker et al. generated single cell transcriptomic and epigenetic datasets from human fetal distal oesophagus and stomach in order to better understand development of this region. They find and provide evidence that a population of distal oesophageal foregut precursor cells are similar to those in observed in Barrett's oesophagus (BO) and argue that BO might be a reversion to fetal-state similar to these cells. These datasets would be valuable to the field and the hypothesis that BO is a fetal conversion is intriguing and impactful.

The revised manuscript is much improved over initial submission. For the most part, my comments have been addressed sufficiently by the authors. I appreciate the difficulty of studying this window of human fetal development. I would recommend acceptance as a Resource Article at Development.

Minor Comments:

Since article is more about BO rather than oesophageal adenocarcinoma similarity to fetal, I would recommend altering the title to reflect this.

While improved, some of the nomenclature is still somewhat confusing, particularly in gene regulatory network section on page 7. In general use the name of cell type with fetal age rather than cluster number if possible.

Include GRHL1 in model in 4H.

Include an x axis in Sup Fig 5B and 7F heatmaps. Is umap missing in Sup Fig 7E top and 9A bottom?

BO typically expresses both gastric and intestinal markers. While fetal oesophagus foregut cells do not have an intestinal-like signature normally, it might be worth a sentence in discussion regarding how fetal epigenetic status could make epithelial cells more competent to activate alternative lineages during adulthood metaplasia.

Reviewer 2

The revised manuscript presents a thoughtful and scientifically rigorous response to the reviewers feedback. The authors have clearly taken the critiques seriously and implemented substantial improvements across the manuscript. The central idea - that Barrett's oesophagus may represent a reactivation of early developmental epithelial programs - is now articulated with greater clarity and nuance. Importantly, the authors avoid overstating their conclusions and acknowledge alternative hypotheses, including the potential involvement of gastric progenitor cells and the role of the microenvironment.

The updated figures and expanded data presentation significantly enhance the transparency and interpretability of the findings. The authors also openly address limitations, including the lack of functional validation in organoid or lineage-tracing systems, and frame these as important avenues for future work.

Overall, the revised manuscript represents a meaningful contribution to the understanding of Barrett's pathogenesis. It offers a valuable developmental resource and proposes a compelling conceptual framework that will inform further mechanistic studies. The study is now well-positioned for publication.

Reviewer 3

SUMMARY OF THE ADVANCE MADE IN THIS PAPER AND ITS POTENTIAL SIGNIFICANCE TO THE FIELD

The revised manuscript has made changes to data presentation and discussion that accurately capture the limitations of the study and discuss the exciting open questions that remain. These include functional testing of how BO results in a reversion to a more developmental esophageal cell state, which, I agree are beyond the scope of the current study. This is a very valuable study, which includes datasets that will be of great use across many fields.

Second revisionAuthor response to reviewers' comments**Reviewer 1:**

The referee is happy with our revised manuscript but outlines some minor issues to improve the clarity of the final version of the manuscript. We have incorporated all of these into the latest revised version:

Since article is more about BO rather than oesophageal adenocarcinoma similarity to fetal, I would recommend altering the title to reflect this.

Response: We have changed the title to "Metaplastic Barrett's oesophagus represents reversion to a developmental-like epithelial cell state" to more accurately reflect the contents as the referee suggests.

While improved, some of the nomenclature is still somewhat confusing, particularly in gene regulatory network section on page 7. In general use the name of cell type with fetal age rather than cluster number if possible.

Response: We have added cell cluster names to the text on page 7 to clarify things further.

Include GRHL1 in model in 4H.

Response: We have added this as requested.

Include an x axis in Sup Fig 5B and 7F heatmaps. Is umap missing in Sup Fig 7E top and 9A bottom?

Response: We added the x-axis to the figures as requested. There were no UMAPs missing from the Supplementary figures but we changed the legend to Sup Fig7E as the positional information (top/bottom rather than left/right) was incorrect.

BO typically expresses both gastric and intestinal markers. While fetal oesophagus foregut cells do not have an intestinal-like signature normally, it might be worth a sentence in discussion regarding how fetal epigenetic status could make epithelial cells more competent to activate alternative lineages during adulthood metaplasia.

Response: We have added a sentence to the discussion to highlight this possibility.

Reviewer 2 and 3:

These reviewers have no further issues to address and are happy for the paper to be published as is.

Third decision letter

MS ID#: dev.204735R2

MS Title: Metaplastic Barrett's oesophagus represents reversion to a developmental-like epithelial cell state

Authors: Syed Baker; Aoibheann Mullan; Rachel Jennings; Karen Piper-Hanley; Yeng Ang; Claire Palles; Neil Hanley; Andrew David Sharrocks
Article Type: Research Article

Dear Dr Sharrocks,

I am happy to tell you that your manuscript has been accepted for publication in Development, pending our standard publication integrity checks.